# Temporal dynamics and microbial interactions shaping the gut resistome in early infancy

Ioanna Chatzigiannidou[1], Pi L. Johansen [1], Rasmus K. Dehli [1], Janne Marie Moll [1], Carsten Eriksen [1], Pernille N. Myers[1], Henrik M. Roager [2], Lili Yang [3], Jakob Stokholm [4,5], Søren J. Sørensen [3], Karen A. Krogfelt [6,7], Martin F. Laursen [8], Urvish Trivedi[3,4], Annika Scheynius[9,10], Karsten Kristiansen [3], Axel Mie[9,11], Johan Alm[9,10] & Susanne Brix [1]✉

Despite the critical role of the gut resistome in spreading of antimicrobial resistance (AMR), strategies to reduce the abundance of antibiotic resistance genes (ARGs) during microbiota development in infancy remain under-explored. Using longitudinal quantitative metagenomic data, we here show that ARGs are present in the gut microbiota from the first week of life, with a peak in absolute ARG abundance and richness at 6 months. Delivery mode significantly affects early ARG dynamics, and vaginally delivered infants exhibit higher ARG abundance due to maternal transmission of *Escherichia coli* strains harbouring extensive resistance repertoires. The abundance of *E. coli* and other ARG-rich taxa inversely correlates with aromatic lactic acid-producing bifidobacteria, and aromatic lactic acids strongly inhibit the in vitro growth of *E. coli* and other opportunistic ARG-rich taxa. Our results highlight temporal and critical microbial interactions shaping the gut resistome in early infancy, pointing to potential interventions to curb AMR during this vulnerable developmental window by promoting colonization of aromatic lactic acid-producing bifidobacteria.

Resistance arises from the expression of specific genes known as antibiotic resistance genes (ARGs), which confer resistance to antibiotics in bacteria[1]. ARGs are ubiquitous in nature and predate human use of antibiotics[2]. Nevertheless, modern antibiotic overuse and misuse have led to an increase in antibiotic resistance and the abundance of ARGs in multiple environments[3-5]. When antibiotic resistance spreads to pathogenic microorganisms, it poses a major public health risk by reducing the efficacy of antibiotics, our primary strategy for

treating, e.g., bacterial infections. For this reason, antibiotic resistance has been characterised by the World Health Organisation (WHO) as 'one of the top public health and development threats'[6].

ARGs are commonly found in the species integral to the human gut microbiota[7]. Collectively, these ARGs form what is known as the gut resistome[8]. ARGs in commensal microorganisms do not pose a direct threat to human health. However, due to the ability of microorganisms to exchange genes through horizontal gene transfer, ARGs

[1]Department of Biotechnology and Biomedicine, Technical University of Denmark, Kgs. Lyngby, Denmark. [2]Department of Nutrition, Exercise and Sports, University of Copenhagen, Frederiksberg C, Denmark. [3]Department of Biology, University of Copenhagen, Copenhagen, Denmark. [4]Copenhagen Prospective Studies on Asthma in Childhood, Herlev and Gentofte Hospital, University of Copenhagen, Gentofte, Denmark. [5]Department of Food Science, University of Copenhagen, Frederiksberg C, Denmark. [6]Department of Bacteria, Parasites and Fungi, Statens Serum Institut, Copenhagen, Denmark. [7]Department of Science and Environment, Roskilde University, Roskilde, Denmark. [8]National Food Institute, Technical University of Denmark, Kgs. Lyngby, Denmark. [9]Department of Clinical Science and Education, Karolinska Institutet, Södersjukhuset, Stockholm, Sweden. [10]Sachs' Children and Youth Hospital, Södersjukhuset, Stockholm, Sweden. [11]Department of Environmental Science, Stockholm University, Stockholm, Sweden. ✉e-mail: sbrix@dtu.dk

in the gut are considered a potential reservoir for the spread of antibiotic resistance to pathogenic microorganisms[9,10]. ARGs have been detected in the human gut microbiota within days after birth[11] and even in infants who have not been exposed to antibiotics[12–14], indicating the initial seeding of the infant gut includes ARG-carrying microorganisms. In fact, the relative abundance of ARGs was reported to be higher in early life than in older toddlers[15,16] and adults[13,17], and is linked to specific bacterial taxa in the gut, such as Gammaproteobacteria, particularly the genus *Escherichia*[13,18].

Understanding how resistance genes are acquired and disseminated in the infant gut is crucial because the resistance gene profile may influence how individuals respond to antibiotic treatments. Previous studies have linked the infant resistome to prior antibiotic exposure[19], delivery mode[14,18], and formula feeding[20], hence pointing to several environmental factors to play a part in the early life gut resistome profile. Nevertheless, many aspects of ARG acquisition and dynamics in early life remain poorly understood; in particular, the interindividual variation in absolute ARG abundance, i.e., the total ARG load, and factors that may modify the ARG load.

In this study, we investigated the temporal infant gut resistome dynamics in the birth cohort ALADDIN (Assessment of Lifestyle and Allergic Disease During Infancy)[21] with access to longitudinally collected faecal samples 8 times from birth up to five years of age from the children and twice from their mothers. By coupling shotgun metagenomic data with absolute quantification of the bacterial load, we assessed the absolute abundance of both the total and the clinically relevant ARGs in mother and infant samples and evaluated the influence of perinatal variables on ARG diversity, prevalence, distribution, and transfer. Genome-resolved metagenomics allowed us to assign ARGs to specific bacterial genomes, identifying key contributors (bacteria and metabolites) to the great interindividual differences in the gut resistome, resulting in experimental validation of microbe-to-metabolite interactions that direct interindividual differences during infancy.

## Results

### Global gut resistome load dynamics during the first five years of life

A total of 313 distinct ARGs were identified across 547 faecal samples collected longitudinally from 56 ALADDIN-enroled children (at 8 time points from 3–6 days to 60 months of age) and their mothers (at third trimester of pregnancy and at 2 months postpartum) with a distribution from 2 to 89 distinct ARGs per sample (Fig. 1a, Supplementary Table 1 and Supplementary Data 1). The majority of ARGs had a low prevalence with 19.5% (61 out of 313) of the ARGs detected in only one sample (Supplementary Fig. 1a). At early time points in the infants (3–6 days, 3 weeks, and 2 months of age), a bimodal distribution in ARG richness was observed, which disappeared by 6 months of age with the detection of a high number of ARGs (average 57) in most infants (Fig. 1a). The bimodal distribution was again observed at 12 months of age, which is similar to a previous report[18], followed by a gradual shifting towards a lower number of ARGs (average 21 at 60 months), comparable to numbers observed in maternal samples during last trimester of pregnancy and at 2 months postpartum (average 20 and 21 ARGs, respectively). The abundance of ARGs relative to the total gene abundance was significantly higher during the first 6 months of life and dropped after 12 months (Fig. 1b). These findings were verified using a read-mapping ARG identification approach (Supplementary Fig 1b), corroborating previous findings[13,15,22]. Additionally, ARG relative abundance was higher at 12 months than at 18 ($P_{adj} = 0.007$), 24 ($P_{adj} = 0.011$), and 60 months ($P_{adj} = 2.4e-5$) of age (Supplementary Table 2), and highly correlated with ARG richness at all ages except for 6 months of age (Supplementary Fig. 1c).

Incorporating information on the absolute ARG abundance in the human gut may provide additional information when studying resistome dynamics, as it enables identification of changes that are not apparent at the relative scale[23]. Hence, to calculate the absolute ARG abundance per sample, we multiplied the percentage of ARG gene reads which was adjusted for gene length with the number of bacteria per gram of faeces in the given sample, determined using flow cytometry (Supplementary Fig. 1d). Absolute ARG abundance was found to vary substantially between individuals during the first 2 months of life and peaked at 6 months, after which it dropped to ~ $1.8 \times 10^6$ ARGs per gram of faeces at 12 months, comparable to the abundance at all following time points and in the mothers (Fig. 1c, Supplementary Table 3). This change over time was also supported when using the read-based ARG mapping method (Supplementary Fig. 1e). Among the ARGs that are annotated as conferring antimicrobial resistance, the subset conferring resistance to clinically relevant antibiotics is more concerning (Supplementary Data 1), such as 3rd generation carbapenems, colistimethate, and tigecycline, which are 'last resort' antibiotics. We further investigated the prevalence and abundance of the subset of clinically relevant ARGs and observed that the absolute abundance of clinically relevant ARGs followed the same pattern as seen for all ARGs across time, although with larger variation amongst infants until 2 months of age (Supplementary Fig. 1f).

Using the Comprehensive Antibiotic Resistance Database (CARD) annotation to infer the mechanism of action of ARGs appearing in early life, it was evident that antibiotic inactivation and antibiotic efflux were the most common resistance mechanisms among detected ARGs when considering their absolute abundance (Fig. 1d). Antibiotic efflux was the most abundant at each time point and antibiotic inactivation had the highest number of unique ARGs across all samples. ARGs that confer resistance against tetracyclines, fluoroquinolones, penams, and cephalosporins were the most common and the most abundant until 6 months of age. ARGs against tetracyclines and fluoroquinolones remained the most common across all ages (Supplementary Fig. 2a). Notably, a high number of ARGs (129 in total, 41% across time points) confer multidrug resistance (Supplementary Fig. 2b). ARG relative and absolute abundance did not differ between antibiotic-naïve infants and those exposed to antibiotics before the 1st sample at 3–6 days of life, or between antibiotic-naïve infants and those who had received at least one course of antibiotics by the age of 1, 1.5 or 2 years regardless of the timing of earlier antibiotics exposure (Supplementary Fig. 3a, b, Supplementary Table 4).

Vaginally delivered infants were found to hold a higher absolute ARG abundance than caesarean delivered infants at 2 months of age (Fig. 1e, Supplementary Fig. 4a), and among the vaginally delivered infants, those born at home had a higher ARG load at 2 and 6 months of age than those delivered at a hospital, also when controlling for covariates (Fig. 1f, Supplementary Fig. 4b). For relative ARG abundances, no significant differences were detected between birth modes (Supplementary Fig. 4c, d). This overall suggests that initial seeding of specific ARG-containing bacteria during vaginal delivery is a major contributor to the absolute ARG abundance in the period between 2 and 6 months of age.

### Microbial composition and birth mode influence ARG diversity

Faecal ARG Shannon diversity generally increased over time during the first months of life, with the highest values at 2 and 6 months of age after which it decreased, and from 24 months of age, the children displayed ARG alpha-diversity levels similar to their mothers (Fig. 2a, Supplementary Table 5). No significant difference in faecal ARG Shannon diversity or ARG richness was recorded between antibiotic-naïve infants and those who received antibiotics before 12 months, 18 months, or 24 months of age (Supplementary Table 6). Infants of mothers who were exposed to antibiotics during pregnancy or during

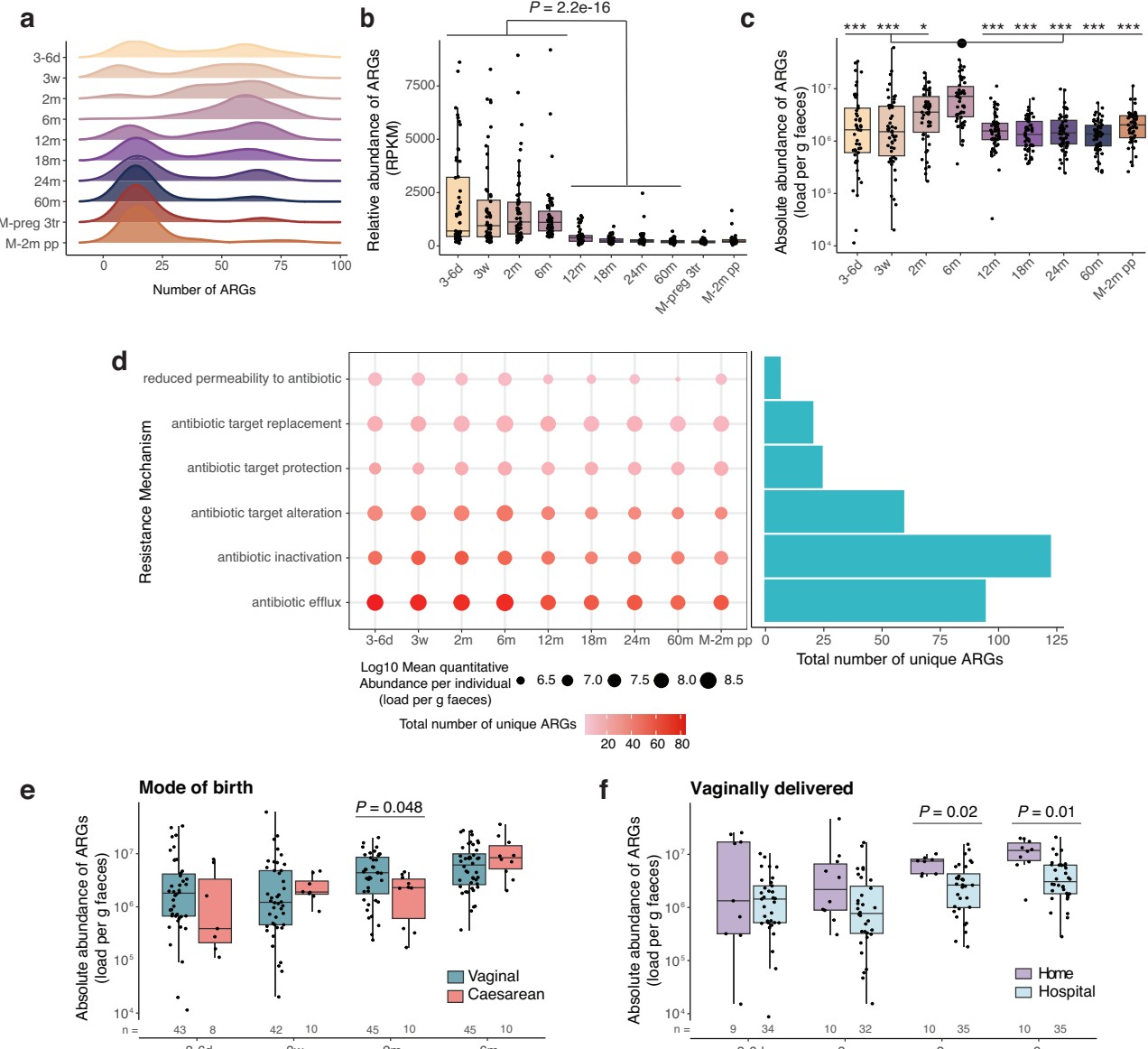

**Fig. 1 | ARG richness and load both peak at 6 months of age and relate to birth mode. a** Distribution of Antibiotic Resistance Gene (ARG) richness in faecal samples at the given time points during early childhood, and in mothers during the 3rd pregnancy trimester (M-preg 3tr), and at 2 months postpartum (M-2m pp). **b** Relative abundance of ARGs in Reads Per Kilobase per Million mapped reads (RPKM) across time points. Samples from 1st week to 6 months were compared with samples from 12 to 60 months with a two-sided Wilcoxon rank-sum test. **c** Absolute ARG abundance, calculated as RPKM/$10^6$ multiplied by bacteria per gram of faeces, displayed per time point. Differences between time points were tested with a two-sided Dunn's test. Stars represent significance compared to the 6 months samples according to the following FDR-adjusted *P* values, *, *P*adj < 0.05, **, *P*adj < 0.01, ***, *P*adj < 0.001. Exact *P*adj values can be found in Supplementary Table 3. **d** Abundance and richness of ARGs ordered by their mechanisms of action.

The gradient colour of the dot corresponds to the number of ARGs with a given resistance mechanism per time point, while the size of the dot represents the average abundance of the given mechanism. The bar plot depicts the total number of ARGs with this mechanism across time points. **e** ARG absolute abundance in early life split by mode of birth and **f**, by home or hospital delivery of vaginally delivered infants. *N* is provided in the figure. Statistics based on linear regression, controlled for sex, family lifestyle and feeding mode. **a**–**d**: 3–6 d (*n* = 51), 3w (*n* = 52), 2 m (*n* = 55), 6 m (*n* = 55), 12 m (*n* = 56), 18 m (*n* = 55), 24 m (*n* = 56), 60 m (*n* = 55), M-preg 3tr (*n* = 56), M-2m pp (*n* = 56). **b**, **c**, **e**, **f** Horizontal lines indicate the median; box boundaries indicate the interquartile range; whiskers represent values within 1.5× the interquartile range of the first and third quartiles. Dots represent the individual data points. Source data are provided as a Source Data file.

delivery at the hospital had a lower faecal ARG Shannon diversity at 6 months (*P* = 0.001), but not at any other time point.

Analysis of the temporal transitioning of the gut resistome composition during early life, assessed using Bray-Curtis dissimilarity, showed significant differences among time points (ANOSIM *R* = 0.337, *P* = 0.001), with an early axis and a late axis splitting at 12 months of age (NMDS1, Fig. 2b). The gut resistome composition was highly correlated with the overall microbiota composition (Mantel test, *r* = 0.6317 and *P* = 0.0001), and with the abundance of specific microbial taxa.

Particularly, *Bifidobacterium spp* and *E. coli*, typical taxa that especially characterise the microbiota in the first 6 months of life (NMDS1, right), were associated with the axis split on the NMDS2. *E. coli* correlated with samples with a higher total ARG relative abundance, and especially a few *Bifidobacterium* species (*Bifidobacterium longum* subsp. *longum* (*B. longum*) and *B. longum* subsp. *infantis* (*B. infantis*)) were dominant in samples with a relatively low ARG relative abundance (Fig. 2b).

Using a permutational multivariate analysis of variance (PERMA-NOVA), birth mode (adjusted for multiple testing, sex and family

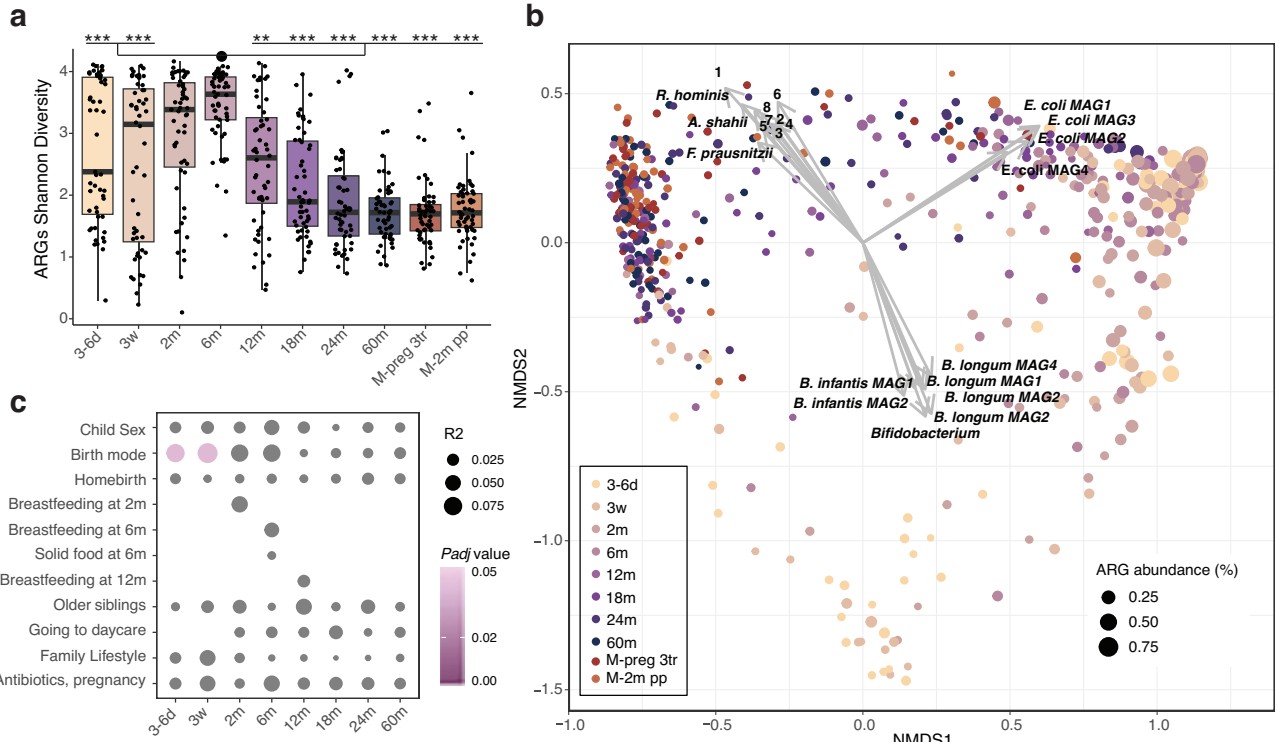

**Fig. 2 | Interindividual ARG diversity in early life. a** ARG Shannon diversity in faecal samples at the given time points during early childhood, 3–6 d (*n* = 51), 3w (*n* = 52), 2 m (*n* = 55), 6 m (*n* = 55), 12 m (*n* = 56), 18 m (*n* = 55), 24 m (*n* = 56), 60 m (*n* = 55), and in mother during the 3rd pregnancy trimester (M-preg 3tr, *n* = 56), and at 2 months postpartum (M-2m pp, *n* = 56). Differences between time points were tested with a two-sided Dunn's test. Stars represent significance compared to the 6 months samples according to the following FDR-adjusted *P* values, *, *Padj* < 0.05, **, *Padj* < 0.01, ***, *Padj* < 0.001. Exact *Padj* values can be found in Supplementary Table 5. Horizontal lines indicate the median; box boundaries indicate the inter-quartile range; whiskers represent values within 1.5× the interquartile range of the first and third quartiles. Dots represent the individual data points. **b** Non-metric multidimensional scaling (NMDS) plot of ARG β-diversity calculated using Bray–Curtis dissimilarity. Individual taxa were fitted on the NMDS plot based on their relative abundance. Taxa with a squared correlation coefficient > 0.15 and a *Padj* < 0.01 are displayed (1: *Alistipes putredinis*, 2–4: *Fusicatenibacter saccharivor-ans*, 5: *Gemmiger qucibialis*, 6: *Odoribacter splanchnicus*, 7: *Alistipes communis*, 8: *Barnesiella intestinihominis*). **c** Perinatal variables associated with ARG β-diversity based on sex, birth mode and family lifestyle adjusted PERMANOVA. Size of the dot depicts the R-squared (R2), and colour depicts the significant FDR-adjusted *P* value (*Padj*). Non-significant conditions are depicted in grey. FDR adjustment was done per time point. *N* is provided in Supplementary Table 7. Source data are provided as a Source Data file.

lifestyle) was found to explain a significant fraction of the ARG β-diversity from 1–3 weeks of age, while no other variables explained a significant amount of variance in ARG β-diversity (Fig. 2c, Supplementary Table 7).

**Only few gut bacterial taxa hold a high number of ARGs**
To decipher which microbial taxa possess the ARGs, we generated metagenome-assembled genomes (MAGs) based on high-quality non-human reads that were binned into 2303 MAGs across all samples. Among these MAGs, we identified 679 MAGs (29.5%) possessing at least one ARG, of which most MAGs (63.3%) contained only a single ARG (Fig. 3a, Supplementary Data 2). The highest number of ARGs per MAG was observed in the four *E. coli* MAGs (with an average of 60 ARGs per genome) followed by MAGs classified as *Klebsiella*, *Citrobacter*, *Enterobacter*, and *Phytobacter* (with 25–36 ARGs per genome), extending findings in earlier studies[16,24]. Most MAGs (87.5%) did not possess any clinically relevant ARGs. The highest number was detected in *Klebsiella pneumoniae*, with seven ARGs, and *Citrobacter freundii*, with six ARGs, while *E. coli* contained four clinically relevant ARGs (Supplementary Fig. 5a).

The number of ARG-containing bacterial taxa (richness) per sample increased until 6 months of age, where it started decreasing, reaching similar values to those of the mothers at 18 months of age (Fig. 3b, Supplementary Table 8). Richness at 6 months was significantly higher than at all other time points except for 2 months of age.

To define whether the ARG-containing MAGs have clinical relevance and are potentially threatening for the individuals, virulence genes were detected in the same genomes (Fig. 3c, Supplementary Data 3). We observed that the four *E. coli* MAGs in the dataset, besides possessing the highest number of ARGs, also possessed the highest number of virulence genes (on average 69 virulence genes per *E. coli* genome), and remained the most abundant among the ARG-rich taxa in the infant gut at all time points (Fig. 3d). The other MAGs with both the highest total ARG richness and the highest number of clinically relevant ARGs (*Klebsiella sp*, *Citrobacter sp*, and *Enterobacter sp*.) har-boured a much lower number of virulence genes (<20 genes per genome), and were also less abundant (Fig. 3d), and less prevalent (Supplementary Fig. 5b) than *E. coli* in the infant gut.

**ARG-rich bacteria are kept in check via aromatic lactic acids**
Given that *E. coli* was identified as the bacterial taxon with both the highest ARG richness, abundance and prevalence in early life, it was not surprising to see that *E. coli* abundance was highly correlated (R = 0.9, P = 2.2e-16) with the ARG relative abundance across time points (Fig. 4a). This implies that strategies to reduce the *E. coli* abundance in the infant gut would have a positive influence on the ARG abundance and potentially reduce the risk of antibiotic resistance spread. We therefore sought to identify factors that may regulate the abundance of *E. coli*.

Birth mode associated with absolute *E. coli* abundance in early life (Supplementary Fig. 6a), with a higher *E. coli* load in vaginally vs.

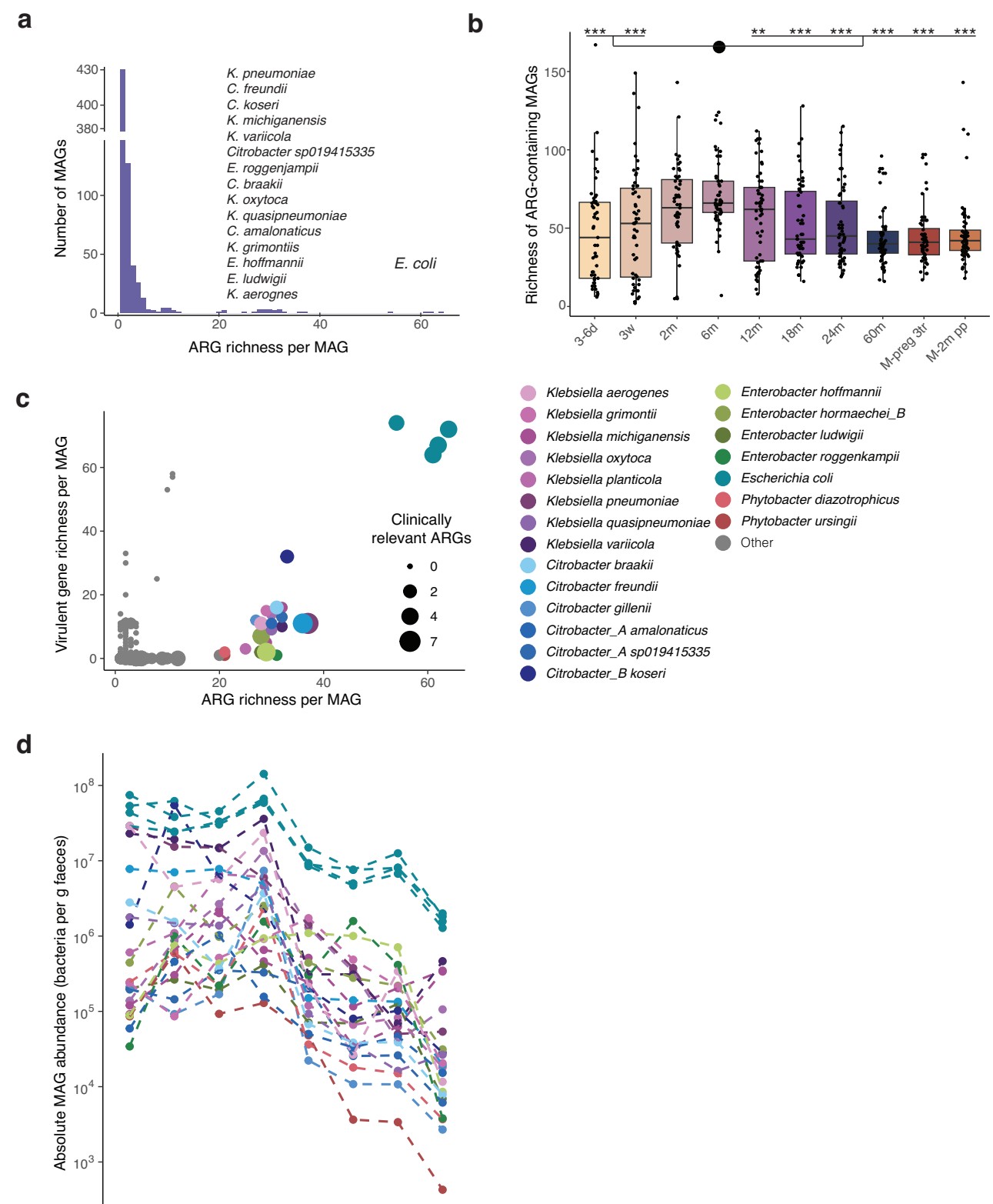

caesarean section delivered infants at 2 months. Furthermore, among the vaginally delivered children, the absolute *E. coli* abundance was significantly higher at 3 weeks and 6 months of age in those born at home.

To investigate a potential direct transfer of *E. coli* from the mother during delivery, we performed a strain similarity analysis for the *E. coli* strains. The strain profile of each sample was defined based on single-

nucleotide variants, and we deemed two samples to share identical *E. coli* strains when the population average nucleotide identity between their *E. coli* strains was higher than 99.999%. Low abundance of *E. coli* in some samples prohibited robust calling of the strain variant, and thus, it was not possible to compare strain similarity between all mother-infant dyads. We were able to compare 13 mother-infant pairs at, at least, one early life time point until 6 months. Six mother–infant

**Fig. 3 | Richness and abundance of ARG-harbouring bacterial species during early life. a** Histogram of ARG richness per metagenome-assembled genome (MAG). **b** Number of ARG-harbouring MAGs in faecal samples at the given time points during early childhood, 3–6 d ($n = 51$), 3w ($n = 52$), 2 m ($n = 55$), 6 m ($n = 55$), 12 m ($n = 56$), 18 m ($n = 55$), 24 m ($n = 56$), 60 m ($n = 55$), and in mothers during the 3rd pregnancy trimester (M-preg 3tr, $n = 56$), and at 2 months postpartum (M-2m pp, $n = 56$). Differences between time points were tested with a two-sided Dunn's test. Stars represent significance compared to the 6 months samples according to the following FDR-adjusted *P* values, *, *Padj* < 0.05, **, *Padj* < 0.01, ***, *Padj* < 0.001. Exact *Padj* values can be found in Supplementary Table 8. Horizontal lines indicate

the median; box boundaries indicate the interquartile range; whiskers represent values within 1.5× the interquartile range of the first and third quartiles. Dots represent the individual data points. **c** ARG and virulence gene richness per MAG. The top 25 MAGs with the highest number of ARGs are coloured according to species name. The size of the dots reflects the number of clinically relevant ARGs per MAG. **d** Absolute abundance of the top 25 MAGs with the highest number of ARGs in faecal samples at the given time points during early life. *N* as in b. Colours refer to species names (see legend for c). Source data are provided as a Source Data file.

pairs shared *E. coli* strains, including all four home deliveries (Fig. 4b). This indicates that vertical transfer of *E. coli* strains from the mother to the child is frequent during vaginal delivery, which is in line with previous findings[25,26].

However, other early-life bacteria may also take part in the observed association with birth mode. As we earlier identified that *E. coli* and specific *Bifidobacterium* MAGs were oppositely associated with ARG composition until 6 months of age, according to axis 2 of the NMDS plot (Fig. 2b), we next focused on the abundance dynamics between *E. coli* and these specific *Bifidobacterium* MAGs until 6 months. Since the four *E. coli* MAGs were found to highly co-occur (Supplementary Fig. 6b), we summed the abundance of *E. coli* for these analyses. The summed relative abundance of *E. coli* correlated inversely with both *B. infantis* and two out of four *B. longum* MAGs at 2 months of age, while not at earlier or later time points, and with the strongest correlation observed for *B. infantis* (Fig. 4c). This trajectory was also observed for the sum of the 4 *Bifidobacterium* MAGs vs. the 4 *E. coli* MAGs (Supplementary Fig. 6c). We also sought to identify if other non-*E. coli* ARG-rich taxa followed the same negative correlation with *Bifidobacterium*. Due to the low prevalence of the non-*E. coli* ARG-rich taxa (Supplementary Fig. 5b), we did not perform the analysis at the individual taxa level but considered the summed abundance of the 21 non-*E. coli* ARG-rich taxa. Inverse correlations between some of the bifidobacteria and the non-*E. coli* ARG-rich taxa were likewise identified at 2 months, as well as at 3 weeks (Supplementary Fig. 6c, d).

It has been suggested that certain bifidobacteria could inhibit the growth of *E. coli* by lowering intestinal pH through the production of lactate and acetate[27], the two main metabolic products of bifido-bacterial carbohydrate metabolism[28], but other metabolites may also play a role, as pH reduction does not seem to be the only mechanism behind the earlier reported growth inhibition[29,30]. Both *B. infantis* and *B. longum* have recently been shown to contain an aromatic lactate dehydrogenase (*aldh*) gene that enables them to convert aromatic amino acids into the aromatic lactic acids (ALAs), phenyllactic acid (PLA), 4-hydroxyphenyllactic acid (4-OH-PLA), and indolelactic acid (ILA)[31]. One of these compounds, PLA has previously been shown to hold antimicrobial activity in the 2.5–20 mg/mL (15–120 mM) range[32,33]. Gene analysis showed that all the above-mentioned *Bifidobacterium* MAGs hold the *aldh* gene (Supplementary Fig. 7a), and we also identified that the summed abundance of the *aldh*+ but not *aldh*- bifidobacteria correlated positively with the faecal levels of the three ALAs (Supplementary Fig. 7b, Supplementary Table 9). The inverse correlation between *E. coli* and *aldh*+ bifidobacterial species could be due to either niche competition or a direct inhibition by bifidobacterial metabolites. When examining the faecal concentrations of the ALAs at 2 and 6 months of age in relation to faecal *E. coli* abundance at the same time points, we found an inverse correlation between *E. coli* abundance and the concentrations of all three ALAs at 2 months of age, but not at 6 months of age (Fig. 4d). The same was seen for the summed abundance of the 21 non-*E. coli* ARG-rich taxa (Supplementary Fig. 8a). Also, the faecal concentration of ALAs correlated inversely with the total (Fig. 4e) and clinically relevant (Supplementary Fig. 8b) ARG relative abundance at 2 months of age, but not at 6 months. The significant inverse correlation between the abundance of *Escherichia* and the

*aldh*+ bifidobacteria *B. infantis*, and *B. bifidum*, and the aromatic lactic acids ILA and 4-OH-PLA were confirmed in the independent cohort Copenhagen Infant Gut (CIG) (Fig. 4f).

To address whether the ALAs can have a direct growth inhibitory effect on *E. coli* at physiologically relevant concentrations, we tested in vitro the effect of *aldh*+ bifidobacteria metabolites on the growth of infant-derived *E. coli* strains with a high genome similarity to the four *E. coli* MAGs in ALADDIN-enrolled children. We identified eleven *E. coli* strains from an infant stool isolate biobank[34,35] with close genomic similarity (ANI range 98.4–99.9%) to the four infant-derived *E. coli* MAGs found in this study. The isolates were grown individually with each of the three ALAs at concentrations similar to those achievable in the infant gut (Supplementary Table 9), as well as the same concentrations of lactate and acetate to compare direct pH-related effects (pKa's within the same range for all except for acetate). No inhibition of *E. coli* growth was seen for lactate or acetate at the tested concentrations, while 10 mM of the three ALAs inhibited the growth in all eleven *E. coli* isolates by up to 125% at 6 h (Fig. 4g, Supplementary Fig. 9a, b). Similar growth inhibition was also observed for infant-derived strains of *Klebsiella pneumoniae* (*K. pneumoniae*) and *Citrobacter freundii* (*C. freundii*), the two ARG-rich taxa that were identified to hold the highest number of clinically relevant ARGs (Fig. 4g, Supplementary Fig. 9a). We further observed a compound- and strain-specific response, with ILA and PLA having longer effects than 4-OH-PLA, and with some of the strains to exhibit growth inhibition of up to 95% at 22 h (Fig. 4g, Supplementary Fig. 9a, b). Altogether, these findings highlight a direct mechanism by which bifidobacteria-derived ALAs may potentially control the abundance of *E. coli* and other ARG-rich opportunistic taxa in early life, and thus, reduce the ARG abundance.

## Discussion

Studying the temporal changes of the infant gut resistome and what drives its composition is relevant in supporting strategies to control the prevalence of resistance genes in the human gut. Using an infant-mother cohort of 56 mother-child dyads with frequent sampling and quantitative measures until five years of age, we discovered quantitative dynamical changes of relevance for the ARG load in the time window from birth until 6 months of age. In particular, this involved specific interactions between *E. coli*, the most dominant and prevalent ARG-rich species in this time window, and ALA-producing bifidobacteria. Faecal concentrations of the three ALAs, PLA, 4-OH-PLA and ILA correlated inversely with relative *E. coli* and ARG abundance in the infant gut, and inhibited the growth of *E. coli* as well as the other ARG-rich species *K. pneumoniae* and *C. freundii* in vitro at physiologically achievable concentrations, suggesting that ALAs may act as potential underlying drivers for the interindividual differences in ARG abundance across infants.

Our study builds upon earlier findings that showed the presence of ARGs in the infant gut since the first week of life, even in antibiotic-naïve infants[12–14] and that ARG relative abundance decreases over time[36]. We added an additional dimension by incorporating information on bacterial load, which revealed a different pattern for absolute ARG abundance compared to what is observed on a relative scale, thereby offering a novel perspective on infant resistome dynamics.

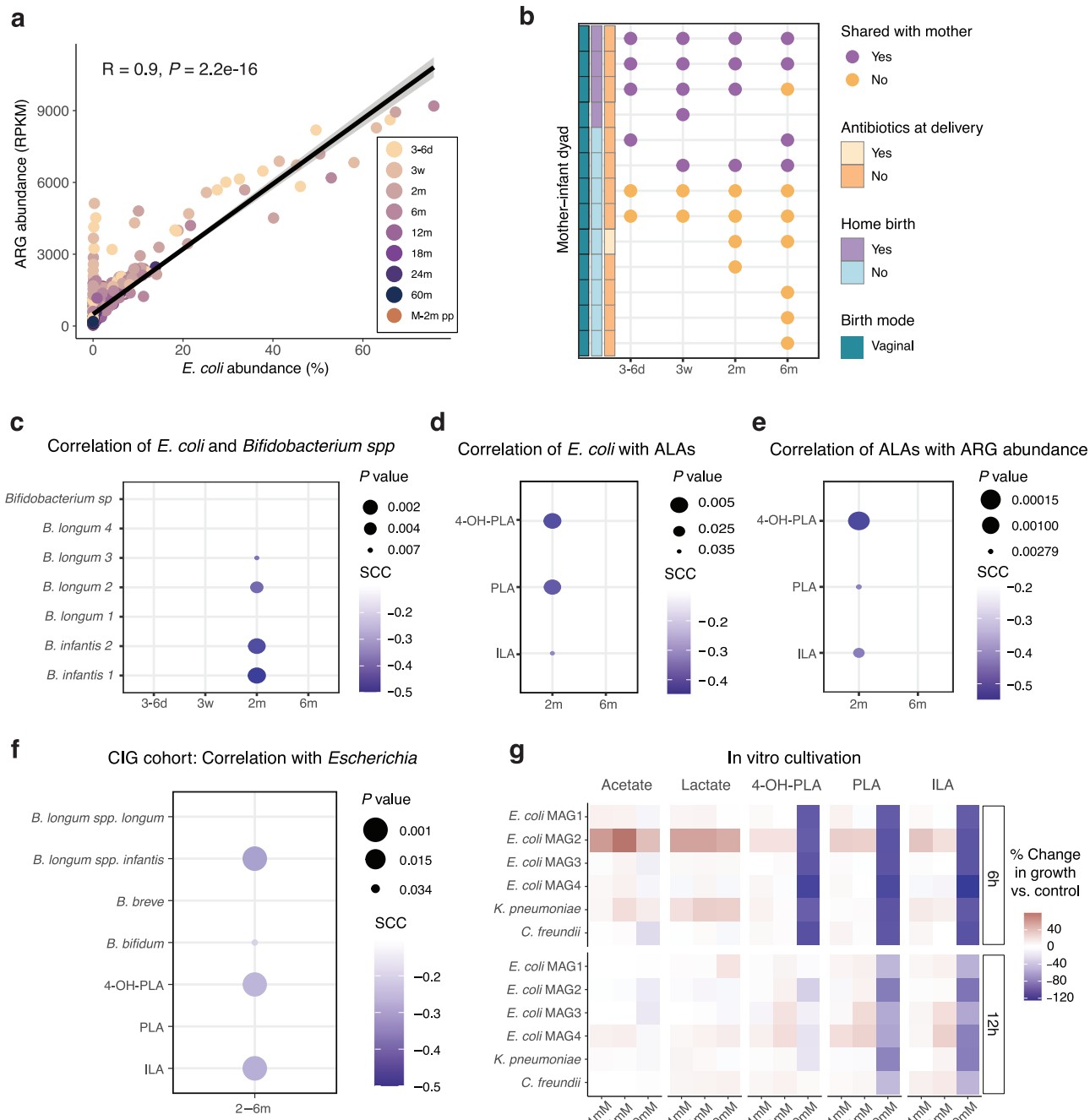

**Fig. 4 | *E. coli* abundance is a driver of total ARG abundance, but growth can be inhibited by aromatic lactic acids. a** Total ARG abundance (RPKM) vs. *E. coli* abundance per sample. The reported *R* and *P* values are based on a two-sided Pearson correlation. The lm regression line (black) and 95% confidence interval (grey ribbon) are displayed. M-2m pp = mothers at 2 months postpartum, *n* as in Fig. 3b. **b** *E. coli* strain sharing between mother-infant dyads during the first 6 months of life. Data are based on faecal samples collected from vaginally delivered infants at the given time points and maternal 3rd trimester pregnancy or 2 months postpartum faecal samples. **c** Two-sided Spearman correlation of abundance of *E. coli* vs. *Bifidobacterium* MAGs associated with ARG composition, 3–6 d (*n* = 51), 3w (*n* = 52), 2 m (*n* = 55), 6 m (*n* = 55). **d, e** Two-sided Spearman correlation of faecal concentrations of the three aromatic lactic acids (ALAs) at 2 and 6 months of age vs. relative abundance of *E. coli* (**d**), and relative ARG abundance (**e**) at the

same time points. *N* = 55 for both time points. Point size represents the *P* value and the point colour corresponds to Spearman's correlation coefficient (SCC). **f** Two-sided Spearman correlation of the relative abundance of *Escherichia* with the absolute abundance of *aldh*+ bifidobacteria and faecal concentrations of the three aromatic lactic acids in 25 infants from the independent Copenhagen Infant Gut (CIG) cohort between 2 and 6 months of age. **g** Growth difference of infant-derived *E. coli*, *K. pneumoniae* and *C. freundii* strains in the presence of three concentrations of each of the metabolites acetate, lactate, phenyllactic acid (PLA), 4-hydroxyphenyllactic acid (4-OH-PLA), and indolelactic acid (ILA) compared to control (no metabolites). One representative strain for each MAG is shown here. The colour of the heatmap represents the average percentage change of three independent biological replicates, each tested in triplicate. Source data are provided as a Source Data file.

Exposure to antibiotics in early life can impact the gut microbiota maturation and the gut resistome composition[12,19,37]. In our cohort, no significant differences were recorded between antibiotic-naïve and antibiotic-exposed infants before the first sample, or at 12, 18 or

24 months of age. It is suggested that antibiotic exposure has an immediate but not long-lasting effect on the gut resistome[12,37], and since the infant gut resistome appears to recover faster than that of the adult[17], this could explain why we did not detect differences, since the

small number of antibiotic-exposed infants and the relative sparsity of the data made it impossible to examine in the identified one-month interval prior to faecal sampling.

Contradicting evidence exists in the literature regarding the effect of birth mode on the gut infant resistome. Some studies report a higher ARG relative abundance in caesarean section than vaginally delivered infants[38]; one identifies it at 1 year of age only[15], and three studies only during the first days of life[11,39,40], while there are also reports of higher ARG relative abundance in vaginally born infants[41] and higher rates of ARG transfer between mothers and infants in vaginally delivered children[42]. No significant differences in ARG relative abundance between birth modes were recorded in our cohort, but we observed a higher ARG absolute abundance in the vaginally born infants at 2 months. These differences between studies could be explained by different cohort characteristics, e.g., resistome differences between countries due to per capita antibiotic usage[43] or country-specific antibiotic prophylaxis for caesarean section. Although evidence on ARG relative abundance with respect to birth mode is inconsistent, a significant correlation between ARG composition and vaginal delivery has been consistently reported[14,15,18], which is also supported by our findings. Literature also suggests a link to breast-feeding vs. formula feeding with the gut resistome[13,20,40], but this was not observed in our study, possibly because of a limited number of formula-fed infants. Due to the temporal shifts observed for the ARG abundance during early life, our study highlights the importance of longitudinal sample collection as well as frequent sampling during the first year when investigating the influence of environmental factors on dynamic changes in ARG abundance during infancy.

We also confirmed that specific bacterial genera were associated with the ARG composition, such as *E. coli*[18,36,44] and several members of the *Bifidobacterium* genus[40,45]. When considering the taxonomy of ARGs, we found that *E. coli* had the highest ARG richness, was most abundant among ARG-rich species in early life and was highly correlated with total ARG abundance across time points. These findings support earlier evidence linking *Escherichia* abundance to the infant gut resistome abundance and composition[13,15,18]. We also reported a higher abundance of *E. coli* in vaginally delivered infants and especially those delivered at home, which was explained by increased vertical transmission from mother to child during birth. This is both supported by our finding that some vaginally born infants shared *E. coli* strains with their mothers and is in agreement with previous findings[25,26,46].

*E. coli* was inversely associated with some *B. longum* and *B. infantis* MAGs, and ALAs, which are metabolic products of the *Aldh* enzyme[31]. We demonstrated in vitro that ALAs at 10 mM can inhibit the growth of *E. coli*, *K. pneumoniae*, and *C. freundii*. With pKa values of ALAs and lactate within the same range, this is indicative of a direct pH-independent mechanism through which ALAs derived from early life *aldh*+ bifidobacteria can inhibit the growth of other early life strains. A pH-mediated inhibition has been shown earlier in respect to acetate and lactate, but was reported to require 5–10× higher concentrations (i.e., 50–100 mM) to directly inhibit *E. coli* growth[47]. Moreover, *E. coli* might even use acetate as a co-substrate to boost its growth at the tested concentrations, depending on the glycolytic flux[48]. Lactate has been demonstrated to prominently permeabilize *E. coli* cells at 5–10 mM[49], and may potentially sensitize *E. coli* cells to make them more susceptible to the antimicrobial effects of the ALAs in the natural gut environment. Due to the probable higher levels of acetate and lactate compared to ALAs in the infant gut[50], and their potential synergistic effect, additional studies are needed to clarify the role of the metabolites and the mechanism behind the inverse correlation with *E. coli* abundance identified in vivo.

Together, our study suggests that promoting colonisation of *aldh*+ bifidobacteria in the infant gut may contribute to maintaining a lower gut resistome abundance by inhibiting the growth of commensal ARG-rich *E. coli* and other opportunistic taxa like *K. pneumoniae*, and *C.*

*freundii* during early life. *Aldh*+ bifidobacteria might be transferred naturally from mother to child during vaginal delivery, or administered as probiotics or as part of a synbiotic. Indeed, administration of pro-biotic *Bifidobacterium* strains to infants has earlier been shown to reduce the abundance of ARGs in stool samples[51,52], however, the route to this effect has so far been unresolved. We here demonstrate a direct antimicrobial activity of all three ALAs against infant-derived ARG-rich *E. coli*, *K. pneumoniae*, and *C. freundii* isolates, but ALAs might act against a range of other ARG-rich species and pathogenic microorganisms, and it would be relevant to investigate this further.

Overall, our study identifies key factors shaping the infant antibiotic resistance profile. We propose that promoting or enhancing ALA production by *aldh*+ bifidobacteria may represent a promising strategy to reduce the prevalence of antimicrobial resistance genes in the infant gut, thereby lowering the risk of resistance dissemination early in life.

## Methods

### ALADDIN study population

The Assessment of Lifestyle and Allergic Disease During Infancy (ALADDIN) birth cohort aims to investigate lifestyle factors in the anthroposophic lifestyle and their association with non-communicable diseases in early life, such as allergy[21]. A total of 466 children were followed from birth until 5 years[53]. Families were categorised into the lifestyle groups, anthroposophic and non-anthroposophic, based on their choice of Maternal Healthcare Centre and parental responses to the questions on a) choice of future type of preschool/school, b) view of lifestyle, and c) the influence of lifestyle on daily life. The families completed detailed questionnaires at enrolment regarding back-ground, health, lifestyle, and diet, and again at 2, 6, 12, 24, and 60 months postpartum, where also health examinations of the children were carried out. Inclusion criteria for this sub-study were based on the existence of longitudinally collected faecal samples from 8 time points during early life of the children and twice from their mothers. Children born pre-term (before gestational week 36) were excluded from the study. Combined this resulted in inclusion of longitudinal samples from 56 mother-child dyads, with children born between 2004 and 2007. Of these, 21 were raised in anthroposophic and 35 in non-anthroposophic families.

Maternal antibiotic use during pregnancy and delivery was assessed by questionnaires at enrolment. Neonatal antibiotic treatment was assessed by maternal interview when children were 2 months of age (one child had received antibiotics intravenously in connection to delivery). Information on antibiotics prescribed to the child was collected from the Swedish Drug Register at the National Board of Health and Welfare, which was launched July 1st 2005. For the seven children born before July 1st 2005, the information on antibiotics use was assessed from interviews and questionnaires performed during early infancy (none of the children had parental report on antibiotics use until the Drug Register was launched). All commercial products reported in the Drug Register were assigned to an antibiotic class based on antibiotic classifications in the CARD.

The study was approved by the Local Ethical Committee Huddinge 2002-01-07 (2002/474-01) and Regional Ethical Committee Stockholm 2010-04-30 (2010/741-32), and written informed consent was obtained from the parents.

### Collection of faecal samples

Faecal samples, in total 547, were collected, frozen within 20 min of collection and stored at −20 °C until later transport in a frozen state for storage at −80 °C.

### Quantification of faecal water metabolites

Aliquots of 100–350 mg from faecal samples collected at 2 and 6 months of age were precisely weighed and diluted 1:4 (w/v) in sterile

Milli-Q water and homogenised by vortexing. An aliquot of each homogenised sample was used for metabolite quantification, while the remaining sample was used for quantifying bacterial load. Sample aliquots were centrifuged at $16,000 \times g$, 4 °C for 5 min. Supernatants were transferred to new tubes and centrifuged at $16,000 \times g$, 4 °C for 10 min. The resulting supernatants were stored at −80 °C in 100 μL aliquots until further sample preparation. Samples were thawed at 4 °C and centrifuged at $16,000 \times g$ for 10 min. Subsequently, 80 μL supernatant was transferred to a new tube in which 20 μL internal standard mix (4 μg/mL isotope-labelled aromatic amino acids: Tyrosine-d4, Phenylalanine-d5, Tryptophan-d5, and Indoleacetic acid-d2) and 240 μL ice-cold acetonitrile were added. Samples were vortexed, incubated at −20 °C for 10 min, and then centrifuged at $16,000 \times g$, 4 °C for 10 min. For each sample, 320 μL supernatant were transferred to a new tube and dried under a stream of nitrogen. Residues were re-dissolved in 80 μL sterile Milli-Q water for a final sample dilution of 1:4 and internal standard concentration of 1 μg/mL, vortexed, and centrifuged at $16,000 \times g$, 4 °C for 5 min. Final supernatants were transferred to liquid chromatography vials with inserts and stored at −20 °C until analysis. 10 μL from 56 random samples (seven random samples from each of the eight time points) were pooled for QC. Targeted UPLC-MS was then employed to quantify the aromatic lactic acids in the faecal water samples using isotope-labelled aromatic amino acids as internal standards as previously described[31].

## Quantification of bacterial load in faeces

The number of bacterial cells was quantified as described in Eriksen et al.[54] with few modifications: Briefly, 100–200 mg faeces was weighed accurately. The homogenised samples, diluted 1:4 in Milli-Q water, were further diluted 2.5-fold in Milli-Q water, and centrifuged at $50 \times g$, 4 °C for 15 min. Supernatant was transferred to a new tube and centrifuged at $8000 \times g$, 4 °C for 5 min. The bacterial pellet was washed twice in sterile PBS + 1% BSA (>98%, Sigma-Aldrich) with centrifugation at $8000 \times g$, 4 °C for 5 min, and resuspended in sterile PBS + 1% BSA. An aliquot was diluted 150x in PBS + 1% BSA + 0.01% Tween20 + 1 mM EDTA, mixed with 1 mM DAPI and BD Liquid Counting Beads (BD Biosciences). 200,000 DAPI+ events were acquired on a FACSCanto II (BD Biosciences) flow cytometer. Bacterial counts were defined by gating the bacterial population on SSC-A/Pacific Blue. The number of recorded count beads was used for calculating the recorded sample volume to determine the number of bacteria per μL solution. The bacterial concentration was multiplied by the dilution factor and divided by the input faecal mass (in grams) to calculate the number of bacteria per g faecal matter.

## DNA extraction of faecal samples

DNA extraction of the 547 faecal samples was conducted in a 96-well format using the Nucleospin 96 soil kit from Macherey Nagel (MN) according to the instructions given by the manufacturer with the following modifications: Aliquots of approximately 50–100 mg were collected from the original faecal samples (kept on dry ice). Samples were mixed with 700 μL buffer SL1 and 150 μL Enhancer SX in MN bead tubes type A containing 0.6–0.8 mm ceramic beads and vortexed on a horizontal vortexer for 5 min at full speed, then centrifuged for 10 min at $11,000 \times g$. 150 μL buffer SL3 was added to each sample and vortexed for 5 s. Samples were stored at −80 °C until the next step. Samples were thawed on ice and vortexed for 5 s, then centrifuged for 10 min at $11,000 \times g$. 800 μL of clear supernatant was transferred to each well of the MN Nucleospin Inhibitor Removal Plate and centrifuged at $4700 \times g$ for 6 min. 250 μL of buffer SB was added to the flow-through in each well of the MN Square-well Block and mixed. 750 μL of the flow-through was transferred onto the MN Nucleospin Soil Binding Plate ($4700 \times g$ for 5 min). Washing of the silica membrane was done according to instructions from the manufacturer, with the exception that all centrifugation steps were done at $4700 \times g$. The plate was air-dried for 15 min and

placed on collection tube strips. 50 μL of preheated (80 °C) buffer SE was loaded directly onto the silica membrane of each well and incubated for 10 min ($4700 \times g$ for 3 min). This step was repeated. The eluted DNA was stored in Eppendorf tubes at −80 °C.

## Shotgun metagenomic sequencing, QC and assembly

Library formation and shotgun sequencing of the 547 faecal samples were performed as previously described using the BGISEQ-500 platform and 100 base pair (bp) paired-end (PE100) sequencing[55]. QC of raw reads was performed using an overall accuracy (OA)-based algorithm to detect and filter low-quality reads using a default parameter (OAfragment = 0.8). After QC, all remaining reads were aligned to the human genome assembly GRCh37 (hg19) to remove human-derived reads. Contigs were assembled from trimmed and host removed reads using MEGAHIT (v1.1.1)[56] (Supplementary Data 4).

## Identification of microbial resistance genes

The contigs were used to predict prokaryotic genes using Prodigal[57] (v2.6.3). Resistance genes were identified by applying Resistance Gene Identifier RGI v5.2.0 and CARD v3.1.2 (2021-04-23)[58] on all prokaryotic predicted genes from all contigs in the 547 faecal samples using standard settings. Genes with a "Perfect" or "Strict" annotation were considered positive hits. Hits flagged as 'nudged' were excluded from further analysis.

A total of 47,960 genes detected in the dataset were assigned as ARGs according to CARD. Annotation of the ARGs on the Antibiotic Resistance Ontology (ARO) revealed that the detected genes belonged to 315 unique functional genes. Prevalence of the genes across samples revealed that three genes were more prevalent that most other genes. Specifically, dfrA42 was present in 99.8% of the samples, adeF was present at 86% of the samples and dfrA43 at 85% of the samples. dfrA42 and dfrA43 genes have been recently described as trimethoprim resistant dihydrofolate reductases[59]. According to CARD, the phylogenetic spread of those genes is limited, and the reported species are not regular members of the gut microbiota. Thus, we considered this ARG to be wrongly annotated, potentially by falsely annotating the regular dihydrofolate reductase as the resistant variant, and the two genes were removed from the ARG dataset. Concerning adeF, previous studies[60] have shown that the current CARD model for the given gene can lead to the wrong annotation of similar RND (Resistance-nodulation-cell division) efflux pump genes as adeF. As these genes also confer antibiotic resistance, it was decided not to discard adeF. The resistance mechanisms and resistance per antibiotic drug classes were assigned based on the RGI annotation. Genes that were annotated to provide resistance against more than one of the drug classes were characterised as multidrug resistance genes. The clinically relevant ARGs were defined based on the latest WHO Bacterial Priority Pathogen List, 2024[61] and Diebold et al.[62].

In addition to the contig-based approach to identify ARGs described above, a read-based approach was also applied. In brief, trimmed host clean pair reads were mapped to CARD, v3.1.2, using the rgi bwt tool from RGI, v6.0.3. Genes with an average coverage ≥ 80% were considered as positive hits. A strong correlation was identified between the ARG abundance determined using the read- vs. the contig-based approach ($R = 0.97$, $P < 2.2e\text{-}16$).

For both the contig- and read-based method, gene abundance was calculated based on RPKM (reads per kilobase per million mapped reads), with the formula: 'reads mapped to ARG gene' / ('gene length in kilobase pairs' × 'reads mapped to all prokaryotic genes'/$10^6$). Absolute ARG abundance was calculated as: (RPKM/$10^6$) × 'bacteria per gram of faeces'.

## Metagenome binning and dereplication

Trimmed metagenomic reads were mapped to assemblies with Bowtie2 v2.5.0[63], and SAM files were converted to BAM format with

SAMtools v1.16[64]. The BAM files were sorted with the samtools sort function and indexed with the samtools index function. SemiBin v1.5.1[65] was run with the single_easy_bin mode using self-supervised learning for the model with the assembly fasta and sorted and indexed BAM file for each sample. The re-clustered output bins of SemiBin for each sample was dereplicated based on 98% average nucleotide identity (ANI) with the dRep dereplicate function of dRep v3.4.0[66] using –S_algorithm ANImf.

The quality of the dereplicated bins was evaluated with checkM2 v1.0.1[67], using the checkm2 predict function with the general model. The MAGs were taxonomically classified using the classify_wf of GTDB-Tk v2.4.0[68] with the GTDB r220. Virulence genes were detected for each genome based on the Virulence Factors of Pathogenic Bacteria (VFDB) database[69] using ABRIcate (v.1.0.1)[70].

For each of the samples, the relative abundance of each MAG with >75% completeness and >5% contamination was calculated with the coverm genome function of coverM v0.6.1[71] using the options --min-read-aligned-percent 0.75, --min-read-percent-identity 0.95, --min-covered-fraction 0.

### Mapping resistance genes to MAGs

An ARG was assigned to a MAG if the contig that contained a certain ARG was part of any medium quality (>50% completeness, <5% contamination) MAG before dereplication. The above resulted in 74% of ARGs, as gene hits, to be mapped to a MAG. ARGs assigned to a MAG accounted for 70% (±22.2) of the ARG relative abundance for each sample. Undereplicated MAGs were matched to their dereplicated representative (98% ANI), which led to a final taxonomic assignment of each ARG.

### Strain analysis

Analysis was performed using InStrain v.1.7.5[72]. First, the MAG catalogue was dereplicated using dRep v3.4.0[66] to 95% ANI similarity to get species level representative genomes. Next, raw reads were mapped to the species-level dereplicated catalogue using bowtie2 v2.4.2[63]. The *E. coli* strain profile of each sample was calculated using the function inStrain profile with the above-created SAM files and the consensus *E. coli* MAG. Finally, inStrain compare was used to compare strain profiles between each two samples. Two samples were considered to share the same profile if they had a coverage of at least 30% and a population average nucleotide identity (popANI) higher than 99.999%. When at least one early life sample fulfilled the above criteria when compared to any of the corresponding maternal samples (pre- or post-partum), it was considered as a shared strain population.

### Aromatic lactate dehydrogenase (*aldh*) gene identification in MAGs

Genome annotation was performed for all MAGs with prokka (v 1.14.5)[73] with default settings. Additionally, 13 genomes of *aldh*+ and *aldh*- bifidobacteria genomes[31] were downloaded from NCBI and annotated similar to the MAGs. All genes annotated as lactate dehydrogenase (*ldh*) (EC number 1.1.1.27) were aligned with multiple sequence alignment using clustal-omega (v 1.2.1)[74] with default settings. A phylogenetic tree was generated based on the alignment with FastTree (v 2.1.11)[75] and was visualised with iTOL v7[76]. All genes that belonged to the same cluster as the *aldh* genes from the annotated reference genomes, and shared the same gene size, were considered as *aldh*, and the MAGs harbouring these genes were considered *aldh*+. All other bifidobacteria strains were designated as *aldh*-.

### Replication cohort (Copenhagen Infant Gut cohort)

The study design, microbiota and metabolomic analyses to determine the ALAs have been described in detail previously[31]. The study was approved by the Data Protection Agency (18/02459), and informed consent was obtained from all parents of infants participating in the study. Briefly, the cohort consists of 25 healthy infants from the Copenhagen region, Denmark. Infant faecal samples were collected approximately every second week between the first week of life and 6 months of age, resulting in a total of 269 samples. As described in detail previously[31], DNA was extracted from faeces, and sequenced using 16S rRNA gene amplicon sequencing. Data processing was performed using the UPARSE pipeline[77]. qPCR and strain- or species-specific primers were used to quantify the abundance of the *aldh*+ bifidobacteria, as described in detail previously[31].

### Growth of infant isolates in the presence of metabolites

All strains used in this study were isolated from faecal samples of different children at 1 week, 1 month or 12 months of age enroled in the Copenhagen Prospective Studies on Asthma in Childhood 2010 (COPSAC2010) cohort, approved by the local Ethics Committee (H-B-2008-093) and the Danish Data Protection Agency (2015-41-3696).

To identify the 3 *E. coli* isolates with closest genomic resemblance to each of the four *E. coli* MAGs identified in the ALADDIN-enroled children, we computed the genomic distance (ANI) between the four *E. coli* MAGs of the dataset and the 690 isolate *E. coli* genomes from COPSAC2010-enroled infants, calculated using Mash[78]. Redundancy amongst the genomes of the 690 isolates was removed by clustering them according to HC1100 scheme provided by EnteroBase[79], including random selection of one representative genome for each cluster. The three closest HC1100 clusters for each of the four MAGs were selected (ANI range 98.4%–99.9%), resulting in the selection of 3 × 4 isolates. One of the isolates was missing in the culture collection, leading to eleven isolates being available for the in vitro cultivation experiment. Three *K. pneumoniae* and three *C. freundii* strains were also selected from the same biobank. One *K. pneumoniae* strain was growing in aggregates, which hindered OD measurements and was therefore removed from the dataset.

The sixteen infant-derived strains were struck from frozen stocks onto LB plates and incubated at 37 °C for 24 h. They were then transferred to LB liquid medium and grown overnight at 37 °C, and finally inoculated at $OD_{600} = 0.1$ into 96-well plates containing diluted LB medium (1:2) and the tested metabolites (DL-indole-3-lactic acid (ILA, I5508, Sigma), DL-p-hydroxyphenyllactic acid (4-OH-PLA, H3253, Sigma), D-phenyl lactic acid (PLA, 376906, Sigma), sodium L-lactate (L7022, Sigma), sodium acetate (S5636, Sigma)) each at a final concentration of 0.1, 1 or 10 mM, or positive controls (1% DMSO and $dH_2O$). ILA, PLA, 4-OH-PLA stocks of 1 M were made in DMSO, and acetate and lactate stocks at 1 M in $dH_2O$. Three different antibiotics (NaI 0.1 mg/mL, Gentamycin 0.01 mg/mL, Tetracyclin 0.01 mg/mL) were used as three separate negative controls and tested in triplicates. The 96-well plates were incubated at 37 °C and $OD_{600}$ was measured every hour for the first 6 h and at 22 h with a Biotek ELx808 plate reader. No growth was observed for any of the negative controls. Growth was estimated as the $OD_{600}$ difference between each sample and the $OD_{600}$ of the negative control samples (average of all replicates for three antibiotic conditions) for the given time point (delta $OD_{600}$ was based on the mean of the triplicates). Three independent experiments were performed with three replicates each time, and the average delta $OD_{600}$ was calculated by using the mean per time to calculate the mean of the three independent replicates.

### Statistics & reproducibility

Statistical analysis was performed using R v4.0.5. No statistical method was used to predetermine sample size. Shannon diversity and richness were calculated using *diversity()* and *specnumber()* functions according to the vegan package (v2.6.4) in R. Differences across time points were calculated with a Dunn's test with the *dunn.test()* function of the R package dunn.test (v1.3.5), with Benjamini–Hochberg correction for multiple comparisons. Effects of environmental variables (delivery mode and home birth) and log10-transformed absolute and relative

ARG abundance were assessed by linear regression models using the *lm*() function from the package stats (v4.2.1). Models were controlled for sex, family lifestyle, and feeding mode at 2 and 6 months. Effects of environmental parameters on absolute *E. coli* abundance were calculated by a two-sided Wilcoxon rank-sum test using *wilcox.test()* from the package stats (v4.2.1).

ARG β-diversity was calculated based on Bray-Curtis dissimilarities calculated from Hellinger-transformed abundance matrices with the R package vegan (v2.6.4). Differences across time points were calculated based on ANOSIM using the *anosim()* function of the same package based on Bray−Curtis dissimilarity on Hellinger-transformed data and 9999 permutations. Correlation between the resistome and microbiota composition was estimated by comparing the Bray−Curtis dissimilarity matrices of the two datasets with the Mantel test using the *mantel()* function of vegan, a Spearman correlation and 9999 permutations. The abundance of the bacterial taxa was fitted to the ARG ordination with the function vegan::*envfit()*. FDR correction of the *P* values of the significantly associated taxa was performed using Benjamini−Hochberg correction, and only the taxa with r > 0.15 and *Padj* value < 0.01 were displayed at the NMDS ordination plot. PERMANOVAs were performed using the *adonis2()* function from vegan with 999 permutations, except for categorical variables with fewer than five children in each stratum. Adonis2() was run with the option by = 'margin', accounting for sex, birth mode, and family lifestyle. FDR correction of the *P* values was performed using Benjamini-Hochberg correction per time point. Correlations between the abundance of *E. coli* and the *Bifidobacterium* species, or *E. coli* and faecal aromatic lactic acids, were calculated by Spearman correlations using the *cor.test()* function from the package stats (v4.2.1). All statistical tests were two-sided, and *Padj*-values were considered significant when lower than 0.05, unless otherwise specified.

### Reporting summary

Further information on research design is available in the Nature Portfolio Reporting Summary linked to this article.

## Data availability

The human clean metagenomic raw sequencing reads and the MAGs generated in this study have been deposited in the sequence read archive (SRA) under accession code BioProject PRJEB84944. The data on individuals' nutritional and lifestyle habits, antibiotic exposure, and faecal aromatic lactic acid concentrations are pseudonymized (coded) personal data, protected and are not available due to data privacy laws. Access can be obtained upon request to johan.alm@ki.se and upon filing of a written agreement that ensures compliance with local legal and ethical regulations. The 16S rRNA gene amplicon sequencing data from the CIG cohort used in this study are available in the SRA under accession code BioProject PRJNA554596. Source data are provided with this paper.

## Code availability

Descriptions of the applied scripts and packages are provided in the methods section.

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

## Acknowledgements

We acknowledge the families participating in the ALADDIN study for their trust and contribution, and the ALADDIN team for their involvement in this work, especially nurse and coordinator Margareta Eriksson, medical doctor Fredrik Stenius, biomedical analyst Carina Wallén, and researcher Catharina Johansson. We would also like to thank Lisbeth Buus Rosholm at DTU Bioengineering for technical support. ALADDIN (J.A., A.M., A.S) was supported by the Swedish Research Council (2012-3011), the Swedish Research Council for Working Life and Social Research (2006-1630), the regional agreement on medical training and clinical research (ALF) between Stockholm County Council and the Karolinska Institutet and the Karolinska University Hospital, the Swedish Asthma and Allergy Research Association, the Swedish Cancer and Allergy Fund, the Ekhaga Foundation, the Frimurare Barnhuset Foundation in Stockholm, and the Hesselman Foundation. Thermo Fisher Scientific, Uppsala, Sweden, provided the study with reagents. H.R. was supported by the Independent Research Fund Denmark (grant 0171-00006B). P.N.M. and S.B were supported by Innovation Fund Denmark (grant 4203-00005B). S.B was the incumbent of the FII Institute Research Chair at DTU in Immune-based Prediction of Disease (2021-2023).

## Author contributions

A.M, A.S, J.A., K.K., and S.B were responsible for the study concept and design. I.C. performed the overall computational analyses. J.M.M. identified the ARGs and performed parts of the computational analysis. I.C. and P.L.J. performed the microbiome and strain analysis. P.N.M. processed the metagenomic sequencing data. R.K.D. and H.M.R. performed the metabolomics analysis. R.K.D. conducted the absolute bacterial quantification, supported by C.E. L.Y. performed the comparison between the *E. coli* MAGs and infant isolates. M.F.L. was responsible for the collection and analysis of the CIG cohort. I.C. and U.T. designed and performed the in vitro *E. coli* experiment. A.S. and J.A. was responsible for the ALADDIN cohort. K.A.K was responsible for the *E. coli* strain isolation, and K.A.K, J.S., and S.J.S for the processing of faecal samples collected in the COPSAC cohort. A.M., A.S., and J.A. obtained the funding and guided the study together with K.K. and S.B. I.C. drafted the original manuscript together with S.B. All authors reviewed and edited the manuscript.

## Competing interests

A.S. was a member of the Joint Steering Committee for the Human Translational Microbiome Programme (CTMR) at Karolinska Institutet, together with Ferring Pharmaceuticals, Switzerland (2016–2023). The remaining authors declare no competing interests.
