## [Transparent Peer Review file · Nature Communications]

Temporal dynamics and microbial interactions shaping the gut resistome in early infancy

Corresponding Author: Professor Susanne Brix

Version 0:

Reviewer comments:

Reviewer #1

(Remarks to the Author)

In an infant-mother cohort of 56 mother-child dyads with frequent sampling and quantitative measures until five years of age, this study investigated the temporal infant gut resistome dynamics in early life. They investigated absolute abundance of ARGs in mother and infant samples and evaluated the influence of antibiotic exposure and perinatal variables on ARG prevalence and distribution.

The noteworthy results are that quantitative dynamical changes of relevance for the ARG load were found in the time window from birth until 6 months of age, involving specific interactions between *E. coli*, the most dominant ARG-rich species, and ALA-producing bifidobacteria. They found no significant differences in absolute ARG abundance and diversity at 12, 18 and 24 months of age between antibiotic-naïve and antibiotic-exposed infants before 12, 18 or 24 months of age, respectively. This study also demonstrated a direct antimicrobial activity of three ALAs against eleven infant-derived *E. coli* isolates.

The paper is well written. Methodology is appropriate in general, but some weaknesses were noted. Details of the power calculation to show that the number of participants used to adequately power each group should be given.

Detailed statistical justification is required to demonstrate that the sample size and the analysis methodologies used are adequate to support the conclusions of this study. Sample size seems low, and the study should be expanded, to confirm the results.

The small number of antibiotic exposed infants and the relative sparsity of the data was a limitation to the conclusions drawn, and a larger data set should be presented to confirm the results.

The abstract lacks sufficient data and information.

This study also demonstrated a direct antimicrobial activity of three ALAs against eleven infant-derived *E. coli* isolates. A number of interesting associations between the gut microbiota, and metabolites were reported. Whether ALAs might act against other ARG rich species in addition to *E. coli* would have strengthened the study findings.

The conclusions rely solely on the analysis of one dataset. Comparative analysis of related datasets is missing, and are required for drawing convincing, solid conclusions.

Furthermore, details of the cohort of mothers and infants are not clearly described. A table with cohort characteristics should be included, including detailed nutritional data of the study participants and antibiotic exposure.

In the discussion, the authors re-write their findings that were previously described in the results section. Instead, the discussion section should provide the readers with the larger implications of the study, provide interpretation to the obtained results, compare the current study to previous related studies, and summarize the research hypothesis and the significance of the findings.

Reviewer #2

(Remarks to the Author)

General comment:

The presented manuscript examines the gut resistome in infancy. In many aspects, the findings recapitulate recent studies that shed light on the natural assembly of infant gut resistome, e.g., PMID 38882494, PMID 39134593, and PMC11135851. An original contribution of this manuscript is the assertion that the abundance of *E. coli* is regulated by metabolites produced by *Bifidobacterium* species, which encode the *aldh* gene for aromatic lactate dehydrogenase. While this is an intriguing hypothesis, the related results are mainly based on correlations and associations. To strengthen the argument for the proposed mechanism, it would be essential to identify the *aldh* gene, establish its direct connection to a *Bifidobacterium* genome, and detect the corresponding metabolites in the same samples.

Besides this limitation, the manuscript is well-written, and the observed results are presented clearly. However, I strongly recommend enhancing the methodological aspects of the resistome and metabolite characterization, as well as providing full transparency in methodology and data reporting. The reported findings seem to be biased due to a lack of a robust methodology.

^

Specific comments:

- The authors choose to use an assembly-based approach to determine the ARGs. However, as low- and even medium-quality reads might be left out in the assembly process, a significant portion of the resistome will not be captured. The limitations of the assembly-based approach have been described earlier (PMID: 39402510, PMID: 36864029). An alternative way to derive the resistome is to map the preprocessed sequencing reads directly to CARD. The validity of conclusions will be more substantial if findings of the alignment- and assembly-based approaches are considered in tandem.
- Figure 1b. Can the authors explicitly explain how the relative abundance of ARGs was calculated? Did the ARG diversity comparisons account for sampling depth?
- Line 108 – How was multidrug resistance defined?
- Line 114 – "This overall suggests that initial seeding of specific ARG-containing bacteria during vaginal delivery is a major contributor to the absolute ARG abundance in the time period between 2 to 6 months of age."  As described in the text now, this statement is not sufficiently backed up by robust statistics and is also in contrast with earlier research (PMID: 36717704). Did the authors control for other factors that significantly influence the gut microbiota and resistome, in particular, the feeding type and antibiotic use? To derive the effect of birth mode on the gut resistome, one could, for example, compare vaginally versus C-section-delivered infants who have the same diet (e.g., all breastfed) and are not exposed to antibiotics. Alternatively, the authors could have used linear mixed-effect modeling to account for the effect of other variables that might be contributing to the effect of birth mode on the gut resistome. The same argument applies to the authors' conclusions on the impact of home births.
- Figure 2a. Some studies have described opposite trends for resistome α -diversity in infancy (e.g., PMID: 37187112). The reported α -diversity results could be influenced by the assembly-based approach, which will negatively impact the samples at the later time points compared to the earlier time points. At later time points, the community is more complex, but because of the limitations of the assembly, this might result in capturing less of the diversity. It would be helpful to describe to what extent the α -diversity of bacterial microbiota and resistome correlates (e.g., Spearman's correlation and Procrustes analysis) as well as show the richness and the evenness separately. This would give a better sense of what is driving resistome α -diversity and strengthen the conclusions.
- Figure 2c. What is the rationale for using PERMANOVA? In general, PERMANOVA is a test of the effect of parallel variables. In contrast, linear mixed effect models consider the combined impact of all factors, can derive the contribution of each variable while considering other potentially modifying factors, and thus seem more appropriate to use.
- Line 175: "This implies that strategies to reduce the *E. coli* abundance in the infant gut would have a positive influence on the ARG abundance and potentially reduce the risk of antibiotic resistance spread."  The authors should also consider that an *E. coli* displacement might not be advantageous. There are likely important ecological roles for *E. coli* in the infant gut, including colonization resistance against other Enterobacteriales with pathogenic potential (e.g., PMID: 38096285).
- Figure 4. Many of the ARGs contributing to the measured resistome are likely endogenous genes that do not have much impact on responses to antibiotic treatment in cases of bacterial infection. To focus on the genes that matter most, the authors could highlight the specific bacterial genes and gene classes identified by WHO (and the literature) as the most critical to clinical antibiotic resistance and antibiotic treatment failure, such as carbapenem and vancomycin resistance genes and third generation cephalosporin-resistance genes. Some of these specific ARGs or ARG classes can be carried by multiple Enterobacteriaceae, not just *E. coli*. This would provide direct clinical relevance to the observations.
- Relatedly, can the authors provide a supplemental table listing the detected ARGs and their classification? When evaluating the risk of ARG carriage, the risk level will differ between different ARGs. A list of detected ARGs will reveal whether the genes are clinically relevant or not. The same is valid for virulence genes – since authors do not disclose the detected genes, the conclusions in the manuscript are based only on authors' statements, without any means for verification by others.
- Figure 4c. Could the authors support the Spearman correlations with a Figure displaying the trajectory of *E. coli* and *Bifidobacterium* spp. abundances over time? The infant feeding type very likely shapes these trajectories; however, this context is missing.
- Line 220 – In what atmosphere was carried out the experiment with growing *E. coli* isolates in the presence of ALAs? Given the anaerobic conditions of the human gut, using anaerobic conditions would be more appropriate, as the inhibition assay might be affected by differences in bacterial metabolism during aerobic conditions.
- Extended Data Table 7 Can the authors show the complete list of metabolite concentrations per sample and not only the medians?
- Did the authors detect the *aldh* gene in the metagenomic data, and to which microbial taxa is it* associated? Without these results, the chapter titled "Early life abundance of *E. coli* is kept in check via *aldh*+ *bifidobacteria* metabolites" and related

text is closer to speculation than a factual finding.

- Was a bead-beating step included during the DNA extraction? As this step is a source of significant technical variation in microbiome studies, I suggest that authors include this information in the manuscript.
- The reported metabolomics method might be biased by the sample processing. First, the raw fecal samples likely had variations in water content, but it is unclear how this was taken into account. Did the authors do any normalization to make sure the same amount of fecal biomass is processed? Secondly, the method does not report any quenching by an organic solvent during fecal sample collection. Fecal samples that are processed without prior quenching will retain enzymatic activities that can alter metabolite levels.
- The authors used flow cytometry to quantify bacterial cells in fecal samples. Did they consider that the results might be affected by the formation of aggregates? Using another method for the enumeration of bacterial cells in parallel, such as qPCR, would increase the reliability of the results.
- Line 365: "the faecal pellet was filtered" – can authors clarify how and why was the fecal pellet filtered?
- Can authors clarify how anthroposophic lifestyle was defined? What do the authors mean by "anthroposophic" and "non-anthroposophic" families?
- Can the authors include the assembly statistics as supplementary information to document the quality?
- As with the resistome and metabolite data, can authors provide the full metadata summarized in Extended Data Table 1? Including all the results according to FAIR data principles will facilitate re-analysis of the dataset and ensure the data integrity.

Version 1:

Reviewer comments:

Reviewer #1

(Remarks to the Author)

Reviewer #2

(Remarks to the Author)

I thank the authors for thoroughly addressing most of my concerns. The manuscript has improved substantially. However, several methodological issues remain.

The manuscript begins with "... known as Antibiotic Resistance Genes (ARGs), which confer resistance to antibiotics in bacteria, archaea, and eukaryotes." It would be helpful if the authors clarify or provide a reference to support the inclusion of eukaryotes in this context.

Regarding the Extended Data Tables, the small text size in the PDF makes them difficult to navigate. Would the authors consider providing these tables in an Excel format to facilitate gene-level exploration and review?

A key methodological concern relates to the approach used to estimate the absolute abundance of ARGs. The method, which involves multiplying the relative abundance of ARGs by bacterial counts per gram of feces, assumes a uniform distribution of ARGs across bacterial cells. However, the authors themselves show that ARGs are concentrated in specific taxa, such as *E. coli*. Furthermore, the approach assumes a linear relationship between the relative abundance of ARGs and total bacterial counts. For example, if bacteria double in number, ARGs will also double in number. Particularly at 6 months, there is likely an increase in bacterial biomass due to the introduction of solid foods.

This raises the possibility that fluctuations in total bacterial load (e.g., around the introduction of solid foods at about 6 months) may not proportionally reflect ARG load, particularly if expanding taxa are ARG-poor. Additionally, could the authors elaborate on the rationale for dividing RPKM values by 10,000 to compute ARG "percentages"? Clarifying this step would help in understanding the conversion from relative to absolute abundances.

On the metabolomics side, it would be valuable to know whether fecal samples were weighed before metabolite quantification. Further clarification on why standards were added at several steps of sample preparation would also enhance reproducibility and understanding of the quantification process.

Finally, the interpretation of the influence of birth mode on the resistome could be further strengthened by addressing discrepancies with earlier studies even more thoroughly. For instance, prior reports, including Busi et al. and a new meta-analysis that utilized 3,981 samples from 1,270 infants (PMID: 40048849), have observed higher resistome loads in caesarean-delivered infants under controlled conditions. A more detailed comparison with those findings, and an acknowledgment of potential cohort differences or sample size limitations, would provide helpful context. The fact that the authors in the presented manuscript did not observe a significant difference in the relative abundance of ARGs between vaginally and caesarean-delivered infants (new Extended Data Fig. 4b) supports the statistically underpowered sample size, with small subgroup sizes that limit statistical inference, as highlighted by reviewer 1.

In conclusion, while the manuscript presents evidence of antibacterial activity by aromatic lactic acids and highlights aldH gene clusters in *Bifidobacterium* MAGs, many of the presented findings have already been discussed in recent studies. The calculation of "ARG absolute abundance" reflects a normalized ARG load per gram of feces rather than a true count of ARGs. The lack of clarification regarding the assumptions underlying the derivation of ARG absolute abundance raises

concerns about the validity of the results, making them uncertain and potentially subject to invalidation in the future. Acknowledging the limitations of this method would enhance the robustness and transparency of the findings.

Dear reviewers,

We hereby submit a revised version of our manuscript NCOMMS-25-03481-T '**Temporal dynamics and microbial interactions shaping the gut resistome in early life**' together with a point-by-point response to the comments of the reviewers.

We would like to thank the reviewers for their insightful comments and suggestions that have contributed to what we consider a much improved version of our manuscript. Following the reviewers' suggestions, we have included data from an independent cohort to support the relevance and broader action of aromatic lactic acids (ALAs), also on bacteria holding clinically relevant ARGs, and we have performed additional experiments to showcase the broader effect of ALA-dependent inhibition on such strains. We have made a thorough editing of the manuscript to address the raised concerns, updating the main figures, the extended data figures, and extended tables as listed below. We have moreover added 11 references.

Fig. 1: 1b, y-axis title edited. 1e,f edited to control for sex, family lifestyle and feeding pattern.

Fig. 2: 2c edited to adjust the PERMANOVA for confounding factors.

Fig. 3: 3c edited to include the number of clinically relevant ARGs. The temporal abundance plot (last plot) has been assigned a d to ease readability, and the dot sizes removed (used to indicate ARG richness) because it is repetitive to information on the x-axis of 3c.

Fig. 4: Addition of replication cohort as new 4f, showing the correlation of *Escherichia* with *Bifidobacterium spp.* and ALAs in the independent CIG cohort. 4g has been updated to include species with highest number of clinically relevant ARGs, *K. pneumoniae* and *C. freundii*, and to display only one representative *E. coli* strain per MAG.

Extended Data Fig. 1: 1b is a new figure representing the relative abundance of ARGs when using the read-based method. 1c is former Fig. 1b. 1d is new and added to show the flow cytometry-based method for bacterial cell quantification. 1e is a new figure of the read-based absolute ARG abundance. 1f is a new figure displaying the absolute ARG-abundance of all ARGs and clinically relevant ARGs.

Extended Data Fig. 2: 2a,b are former Extended Data Fig. 1c,d.

Extended Data Fig. 3: 3a,b are former Extended Data Fig. 2a,b. Data displayed in former Extended Data Fig. 2c-h are now in Extended Data Table 5.

Extended Data Fig. 4: 4a is a new figure displaying the effect size from linear regression models of data in Fig. 1e,f. 4b,c,d are new figures displaying the effect size and the relative abundance of ARGs separated by mode of birth or location of vaginal delivery.

Extended Data Fig. 5: 5a is a new figure displaying the distribution of clinically relevant ARGs across bacterial genomes. 5b is a new figure displaying the prevalence of ARG-rich taxa across time points.

Extended Data Fig. 6: 6a is former Extended Data Fig. 4a. 6b is a new figure displaying the co-correlation between the four *E. coli* MAGs. 6c is a new figure displaying the average relative abundance of the four *E. coli* MAGs, the 4 *Bifidobacterium* MAGs (*B. infantis* MAG1, *B. infantis* MAG2, *B. longum* MAG2, *B. longum* MAG3) and 21 non-*E. coli* ARG-rich taxa. 6d is a new figure displaying the correlation between *Bifidobacterium* and the 21 non-*E. coli* ARG-rich taxa.

Extended Data Fig. 7: 7a is a new figure with the phylogenetic tree of the *ldh* gene. 7b is a new figure displaying the correlation between *aldh+* and *aldh-* bifidobacteria and the faecal levels of ALAs.

Extended Data Fig. 8: 8a,b are new figures displaying the correlation between ALAs and the 21 non-*E. coli* ARG-rich taxa and the relative abundance of the clinically relevant ARGs, respectively.

Extended Data Fig. 9: 9a includes growth inhibition data of the additional *E. coli* isolates previously shown in Fig. 4f, as well as new data of the growth inhibition of additional strains of *K. pneumoniae* and *C. freundii*. 9b is an edited version of former Extended Data Fig. 4b with addition of the growth curves of the 2 *K. pneumoniae* and 3 *C. freundii* strains.

Extended Data Table 1: updated version of former Extended Data Table 1

Extended Data Table 2: new table listing all ARGs detected in the dataset

Extended Data Table 3 is former Extended Data Table 2

Extended Data Table 4 is former Extended Data Table 3

Extended Data Table 5: new table with listing data formerly presented in Extended Data Fig. 2c-h as well as new data.

Extended Data Table 6 is former Extended Data Table 4

Extended Data Table 7: new table with listing data formerly presented in Extended Data Fig. 3 as well as new data.

Extended Data Table 8 is an updated version of former Extended Data Table 5

Extended Data Table 9: new table with ARG annotation to Metagenome Assembled Genomes

Extended Data Table 10 is former Extended Data Table 6

Extended Data Table 11: new table with virulence gene annotation to Metagenome Assembled Genomes

Extended Data Table 12 is former Extended Data Table 7

Extended Data Table 13: new table listing the assembly quality metrics

Changes in the manuscript file are marked as red (track changes). In the below, the manuscript line indications refer to the marked up version.

Please find below our point-by-point answers **in blue font** to the reviewers' comments and new text marked in *italic*:

Reviewer #1 (Remarks to the Author):

In an infant-mother cohort of 56 mother-child dyads with frequent sampling and quantitative measures until five years of age, this study investigated the temporal infant gut resistome dynamics in early life. They investigated absolute abundance of ARGs in mother and infant samples and evaluated the influence of antibiotic exposure and perinatal variables on ARG prevalence and distribution.

The noteworthy results are that quantitative dynamical changes of relevance for the ARG load were found in the time window from birth until 6 months of age, involving specific interactions between *E. coli*, the most dominant ARG-rich species, and ALA-producing bifidobacteria. They found no significant differences in absolute ARG abundance and diversity at 12, 18 and 24 months of age between antibiotic-naïve and

antibiotic-exposed infants before 12, 18 or 24 months of age, respectively. This study also demonstrated a direct antimicrobial activity of three ALAs against eleven infant-derived *E. coli* isolates.

The paper is well written. Methodology is appropriate in general, but some weaknesses were noted. Details of the power calculation to show that the number of participants used to adequately power each group should be given.

R1-Q1: Detailed statistical justification is required to demonstrate that the sample size and the analysis methodologies used are adequate to support the conclusions of this study. Sample size seems low, and the study should be expanded, to confirm the results.

Answer: We acknowledge the reviewer's relevant comment concerning the sample size of our study and we recognize that the number of 56 individual children might appear small. Nevertheless, our cohort is characterized by a frequent longitudinal sampling at 8 time points from birth until 5 years of age resulting in 457 faecal samples from children across the time points + two time points from their mothers (128 samples in total). For replication of findings related to the absolute ARG dynamics in early life, one would need at least 4 sampling time points between birth and 12 months incl. one around 6 months. The number of individuals in our cohort is comparable or larger than similar infant cohorts with frequent sampling during the first year, and shotgun metagenomics sequencing (please see ID no. 1-3 in below table providing an overview of publicly available infant cohorts with frequent longitudinal sampling between 1 month and 1 year).

ID no.	Publication	PMID	Shotgun (S)/ Amplicon (A)	Total bacteria quant.	# of children	Sampling period	Sampling period range
Cohorts of full-term infants with at least 4 time points between 1-12 months and one at 6 months							
1	Busi 2021	36717704	S	no	20	5d, 1m, 6m, 12m	5d-12m
2	Trosvik 2024	39134593	S	no	12	monthly intervals	0-12m
3	Baumann-Dudenhoeffer 2018	30374198	S	no	60	0m, 1m, 2m, 3m, 4m, 5m, 6m, 7m, 8m	0-8m
4	Samarra 2025	40603287	S	no	66	7d, 1m, 6m, 12m	1w-12m
5	Jokela 2022	36174236	A	yes, in 92	144	1d, 2d, 1w, 4w, 6w, 12w, 6m, 9m, 12m	1d-12m
Cohorts of full-term infants with other sampling intervals							
6	Backhed 2015	25974306	S	no	98	newborn, 4m, 12m	0-12m
7	Shao 2019	31534227	S	no	596	4d, 7d, 21d, infancy (4-14m)	0-14m
8	Barheet 2023	37187112	S	no	72	7d, 28d, 120d, 365d	7d-12m
9	Vatanen 2016	27133167	S	no	222	0-3y*	0-3y
10	Dai 2023	25405552	S	no	1424	3m, 12m	3m-12m
11	D-Souza 2020	31832638	S	no	63	6w, 4m, 6m	6w-6m
12	Yassour 2018	30001517	S	no	44	0d, 14d, 1m, 2m, 3m	0-3m
13	Ferreti 2018	30001516	S	no	25	1d, 3d, 7d, 1m, 4m	0-4m
14	Pärnänen 2018	30250208	S	no	16	1m, 6m	1m-6m
15	Hickman 2024	39333099	A	no	967	3w, 6w, 3m, 9m, 12m, 18m, 24m	0-24m
16	de Muinck 2018	29884786	A	no	12	almost every day for a year	0d-1y

*Sampled every month but not sequenced every month for all individuals, 1(25), 2(31), 3(63), 4(35), 5(20), 6(7), 7(8), 9(4), 10(2), 11(2), 12(2), 13(2)

The families in the ALADDIN birth cohort are categorized into three lifestyle groups: anthroposophic, partly anthroposophic, and non-anthroposophic (<https://pubmed.ncbi.nlm.nih.gov/21651566/>). In our present study we included families with anthroposophic and non-anthroposophic lifestyle where the children had longitudinal faecal samples from all 8 time points during early life and mothers had both samples. Due to the lack of faecal samples across time points from other study participants and our decision not to include families with partly anthroposophic lifestyle, we are unfortunately unable to include more samples from the same cohort to support the study results.

Since our cohort data nicely replicates the recent reports combining publicly available data, (<https://pubmed.ncbi.nlm.nih.gov/38217470/>; <https://pubmed.ncbi.nlm.nih.gov/38882494/>) of a decrease over time of the relative ARG abundance, with strong statistical support ($P < 2.2e-16$, Fig. 1b), we would argue that the cohort size is large enough to robustly capture dynamical changes in absolute ARG abundance, which was one of two main findings of our study. The calculation of absolute ARG abundance relies on the generation of quantitative profiles. As recently acknowledged in the article by Nishijima et al., Cell 2025 (<https://pubmed.ncbi.nlm.nih.gov/39541968/>), the quantitative profiles are both “labor-intensive, costly, and impractical for large-scale microbiome studies. Hence, the vast majority of public or ongoing metagenomic studies do not take into account associated microbial loads in their analyses.” Their article focused mainly on adults, with a few samples collected in children above 3 years of age, and there are presently no infant cohorts with shotgun metagenomics and longitudinal sampling that includes quantitative profiles, as listed in the above table. In Nishijima et al. Cell 2025 they develop an algorithm for prediction of quantitative profiles, but this has not been trained on early life samples, where the dynamics are rapidly changing, and when testing it we find that it does not provide near accurate predictions when using it on the data set from our study. We have therefore decided not to try to replicate our findings using predicted quantitative profiles in this age group.

It would be possible for us to perform the quantitative analysis in another cohort, given that we will be able to get access to faecal samples in one of the above listed cohorts (no. 3 or 5), if it is deemed necessary by the reviewers and editor to justify our findings regarding the absolute ARG dynamics in early life. Unfortunately, this was not possible within the time frame for this revision, and will delay the publication by at least another 6-9 months due to agreements, sample shipment and processing.

To provide a comparative analysis of the other major study finding on the relation between early-life *E. coli*, *aldh+* bifidobacteria and ALAs, we have now been able to include data from an independent cohort, the Copenhagen Infant Gut (CIG) cohort (<https://pubmed.ncbi.nlm.nih.gov/34675385/>). In the CIG cohort, children were frequently sampled from birth up to 6 months of life, the gut microbiota was characterized using 16S rRNA gene amplicon sequencing, *aldh+* bifidobacteria were quantified using qPCR, and the three ALAs were quantified in infant faecal samples at the same time points. We confirmed in this independent cohort that in infants between 2 and 6 months, the relative abundance of *Escherichia* was inversely correlated with the sum of ALAs, ILA and 4-OH-PLA (now included as new Fig. 4f, see below). The identified inverse correlation of *Escherichia* with the different *aldh+* bifidobacteria species was also confirmed in the CIG cohort.

We have inserted I256- ‘*The significant inverse correlation between the abundance of Escherichia and the aldh+ bifidobacteria B. longum spp. infantis, and B. bifidum, and the aromatic lactic acids ILA and 4-OH-PLA were confirmed in the independent cohort Copenhagen Infant Gut (CIG) (Fig. 4f).*’

Information on the CIG cohort and the performed analyses was added to the methods section, I560-, ‘*The study design, microbiota and metabolomic analyses to determine the ALAs have been described in detail previously³⁰. The study was approved by the Data Protection Agency (18/02459), and informed consent was obtained from all parents of infants participating in the study. Briefly, the cohort consists of 25 healthy infants from the Copenhagen region, Denmark. Infant faecal samples were collected approximately every*

second week between the first week of life and 6 months of age, resulting in a total of 269 samples. As described in detail previously³⁰, DNA was extracted from faeces, and sequenced with 16S rRNA gene amplicon sequencing. Data processing was performed using the UPARSE pipeline⁷². qPCR and strain- or species-specific primers were used to quantify the abundance of the *aldh+* bifidobacteria, as described in detail previously³⁰.

New Fig. 4f:

Figure text for new Fig. 4f: Spearman correlation of the relative abundance of *Escherichia* with the absolute abundance of *aldh+* bifidobacteria and faecal concentrations of the three aromatic lactic acids in 25 infants from the independent CIG cohort between 2 and 6 months.

R1-Q2: The small number of antibiotic exposed infants and the relative sparsity of the data was a limitation to the conclusions drawn, and a larger data set should be presented to confirm the results.

Answer: We agree with the reviewer that the sparsity of antibiotic consumption within the cohort does not allow for specific conclusions to be drawn in relation to the effect of antibiotic intake on ARG dynamics during infancy in this study. We have acknowledged this in our manuscript in I300-303 that read: ‘... *this could explain why we did not detect differences, since the small number of antibiotic exposed infants and the relative sparsity of the data made it impossible to examine in the identified one-month interval prior to faecal sampling*’

Although the effect of antibiotic consumption was not a main focus of our study (included as Extended Data Table 5 and 7), we did make an effort and examined whether it is possible to follow the reviewer’s suggestion to include a larger dataset to investigate this. However, even after thorough searches, we were unable to locate an existing dataset with longitudinal sampling beyond 8 months of age that also provides detailed information on antibiotic exposure (please refer to the inserted table below, listing the same longitudinal cohorts with frequent sampling as in R1-Q1). Since the ARG Shannon diversity is only gradually starting to decline at 12 months of age (Fig. 2a), we decided it would not add additional information if we examined the age-dependent time to effect of antibiotics (abx) exposure on ARG diversity, as compared to already existing reports (referred to in our discussion, see below).

ID no.	Publication	PMID	Country of origin	To examine effect of antibiotics exposure on infant ARG diversity
Cohorts of full-term infants with at least 4 time points between 1-12 months and one at 6 months				
1	Busi 2021	36717704	Luxembourg	NO: only 20 individuals. Has abx info, however only 3 with abx
2	Trosvik 2024	39134593	Norway	NO: only 12 individuals, only one child with abx
3	Baumann-Dudenhoefter 2018	30374198	USA	NO: 9 time points including 6m, but last at 8m, info on abx
4	Samarra 2025	40603287	Spain	NO: 66 infants, no abx info provided, only 7 infants with abx 12m
5	Jokela 2022	36174236	Finland	NO: amplicon seq (AMR gene calling not possible)
Cohorts of full-term infants with other sampling intervals				
6	Backhed 2015	25974306	Sweden	NO: no 6m time point, only 3 time points
7	Shao 2019	31534227	UK	NO: one sample during infancy for each of the 302 infants taken at 8.75 ± 1.98 m
8	Barheet 2023	37187112	Norway	NO: no abx in full terms. No 6m time point
9	Vatanen 2016	27133167	Finland, Estonia, Russia	NO: info on abx exposure, and many individuals, only 1 with paired samples at 2m and 6m
10	Dai 2023	25405552	Canada	NO: only 2 time points
11	D-Souza 2020	31832638	South Africa	NO: only 3 time points, only prophylactic cotromoxazole treated infants, no other abx
12	Yassour 2018	30001517	Finland	NO: no info on abx, last time point at 3m
13	Ferreti 2018	30001516	Italy	NO: no time points beyond 4m
14	Pärnänen 2018	30250208	Finland	NO: no abx use in cohort
15	Hickman 2024	39333099	Finland	NO: amplicon seq, AMR gene calling not possible
16	de Muinck 2018	29884786	Norway	NO: 12 children, 1 with abx, amplicon seq

We decided to additionally investigate whether exposure to antibiotics before the first faecal sample has an effect on the infant gut resistome. However, we did not observe significant differences either, except for ARG Shannon diversity at 6 months that was lower for infants exposed to antibiotics before the 1st sample in week 1 as compared to antibiotic naïve infants. We have incorporated the results in Extended Data Table 5 and 7 and updated the manuscript in I125- *‘those exposed to antibiotics before the 1st sample at 3-6 days of life, or between antibiotic-naïve infants and... (Extended Table 5)’* and I147-, referring to Extended Table 7: *‘Infants of mothers that were exposed to antibiotics during pregnancy or during delivery at the hospital had a lower faecal ARG Shannon diversity at 6 months ($P = 0.001$), but not at any other time point.’*

We would like to emphasize that we report that in I145-: *‘No significant difference in faecal ARG Shannon diversity or ARG richness was recorded between antibiotic-naïve infants and those who received antibiotics before 12 months, 18 months, or 24 months of age (Extended Data Table 7)’*. We refer to studies that have identified specific relationships between antibiotics exposure and the resistome. This information can be found in I277-: *‘In our cohort, no significant differences were recorded between antibiotic-naïve and antibiotic-exposed infants before the first sample, or at 12, 18 or 24 months of age. It is suggested that antibiotic exposure has an immediate but not long-lasting effect on the gut resistome^{11,36}, and since the infant gut resistome appears to recover faster than that of the adult¹⁶, this could explain why we did not detect differences, since the small number of antibiotic exposed infants and the relative sparsity of the data made it impossible to examine in the identified one-month interval prior to faecal sampling.’*

R1-Q3: The abstract lacks sufficient data and information.

Answer: We acknowledge the request to provide additional information, but adhering to the 150-word limit makes it inherently difficult to add many details.

The abstract now reads:

'Despite the critical role of the gut resistome in spreading of antimicrobial resistance (AMR), strategies to reduce the abundance of antibiotic resistance genes (ARGs) during microbiota development in infancy remain underexplored. Using longitudinal quantitative metagenomic data, we here show that ARGs are present in the gut microbiota from the first week of life, with a peak in absolute ARG abundance and richness at 6 months. Delivery mode significantly affects early ARG dynamics, and vaginally delivered infants exhibit higher ARG abundance due to maternal transmission of Escherichia coli strains harbouring extensive resistance repertoires. The abundance of E. coli and other ARG-rich taxa inversely correlates with aromatic lactic acid-producing bifidobacteria, and aromatic lactic acids strongly inhibit the in vitro growth of E. coli and other opportunistic ARG-rich taxa. Our results highlight temporal and critical microbial interactions shaping the gut resistome in early infancy, pointing to potential interventions to curb AMR during this vulnerable developmental window by promoting colonization of aromatic lactic acid-producing bifidobacteria.'

R1-Q4: This study also demonstrated a direct antimicrobial activity of three ALAs against eleven infant-derived *E. coli* isolates. A number of interesting associations between the gut microbiota, and metabolites were reported. Whether ALAs might act against other ARG rich species in addition to *E. coli* would have strengthened the study findings.

Answer: In accordance with the reviewer's suggestion, we have now performed additional experiments. Based on the comparative analysis of total and clinically relevant ARGs (please refer to R2-Q10), we have now incorporated *in vitro* inhibition data for two additional bacterial species, *Klebsiella pneumoniae* and *Citrobacter freundii*, as they were determined to hold the highest number of total ARGs after *E. coli* and to harbour the highest number of clinically relevant ARGs. We cultivated two different *Klebsiella pneumoniae* and three different *Citrobacter freundii* strains that were isolated from infant faeces and belonged to the same biobank as the *E. coli* strains. The experiments were performed as three independent repetitions (different days), and in technical triplicates, as with the *E. coli* isolates. As visualized in new Fig. 4g and Extended Data Fig. 10a (inserted below), we observed inhibition patterns similar to those for *E. coli*. This implies that the inhibitory effect of ALAs is not specific for *E. coli* but also translates to other ARG-rich Gram-negative pathobionts. We have inserted I260-262 '*Similar growth inhibition was also observed for infant-derived strains of K. pneumoniae and C. freundii, the two ARG-rich taxa that were identified to hold the highest number of clinically relevant ARGs (Fig 4g, Extended Data Fig. 9a)*' and have updated Fig. 4g. For simplicity, only one of the strains for each species was included in Fig. 4g, and the data for the remaining ones (showing similar results) are available in Extended Data Fig. 9a.

Updated Fig. 4g:

Figure text for updated 4g: Growth difference of infant-derived *E. coli*, *K. pneumoniae* and *C. freundii* strains in the presence of three concentrations of each of the metabolites acetate, lactate, PLA, 4-OH-PLA, and ILA compared to control (no metabolites). One representative strain for each MAG is shown here. The colour of the heatmap represents the average percentage change of three independent biological replicates, each tested in triplicates.

Updated Extended Data Fig. 9a:

Figure text for Extended Data Fig. 9a: a, Growth difference in the presence of three concentrations of metabolites (acetate, lactate, 4-OH-PLA, PLA, ILA) compared to no metabolites for seven additional *E. coli*, one *K. pneumoniae* and two *C. freundii* that are not displayed in Fig 4g. The colour of the heatmap represents the average percentage change of three independent biological replicates, each tested in triplicates.

The detailed growth curves are included in Extended Data Fig. 9b (see below) and the methods section 1583- was updated to provide information on the additional strains: 'Three *K. pneumoniae* and three *C. freundii* strains were also selected from the same biobank. One *K. pneumoniae* strain was growing in aggregates, which hindered OD measurements and was therefore removed from the dataset.'

Updated Extended Data Fig. 9b:

Figure text for 9b: b, Growth curves of the eleven *E. coli*, two *K. pneumoniae* and three *C. freundii* strains grown under different concentrations of lactate, acetate, ILA, PLA and 4-OH-PLA. Delta OD represents the OD600 for individual cultures after subtracting the OD600 negative controls (same isolate but treated with antibiotics). The line represents the average of the three independent biological replicates and the ribbon depicts the standard deviation.

R1-Q5: The conclusions rely solely on the analysis of one dataset. Comparative analysis of related datasets is missing, and are required for drawing convincing, solid conclusions.

Answer: We agree that it is critical to perform comparative analysis of related datasets. Please refer to R1-Q1 where we explain the added additional analyses.

R1-Q6: Furthermore, details of the cohort of mothers and infants are not clearly described. A table with cohort characteristics should be included, including detailed nutritional data of the study participants and antibiotic exposure.

Answer: Cohort characteristics are presented as Extended Data Table 1, but we realized that information on pets and exposure to farm animals was missing. This has now been added.

R1-Q7: In the discussion, the authors re-write their findings that were previously described in the results section. Instead, the discussion section should provide the readers with the larger implications of the study, provide interpretation to the obtained results, compare the current study to previous related studies, and summarize the research hypothesis and the significance of the findings.

Answer: According to the reviewer's suggestion, we have edited our discussion to remove the repetitive parts focusing on the interpretation of our findings and their significance. Please refer to the track changes in the discussion to see the substantial number of changes.

Reviewer #2 (Remarks to the Author):

General comment:

R2-Q1: The presented manuscript examines the gut resistome in infancy. In many aspects, the findings recapitulate recent studies that shed light on the natural assembly of infant gut resistome, e.g., PMID 38882494, PMID 39134593, and PMC11135851. An original contribution of this manuscript is the assertion that the abundance of *E. coli* is regulated by metabolites produced by *Bifidobacterium* species, which encode the *aldh* gene for aromatic lactate dehydrogenase. While this is an intriguing hypothesis, the related results are mainly based on correlations and associations. To strengthen the argument for the proposed mechanism, it would be essential to identify the *aldh* gene, establish its direct connection to a *Bifidobacterium* genome, and detect the corresponding metabolites in the same samples.

Answer: As it is highlighted by the reviewer, our study corroborates earlier studies that describe the infant gut resistome, but we would argue that our study provides additional results including but not limited to the role of ALAs as a potential way to regulate the abundance of ARG-rich species, including *E. coli*. The first two studies mentioned above <https://pubmed.ncbi.nlm.nih.gov/38882494/>, and <https://pubmed.ncbi.nlm.nih.gov/39134593/> describe the decreasing ARG relative abundance overtime, which confirms earlier findings in <https://pubmed.ncbi.nlm.nih.gov/30250208/>, and <https://pubmed.ncbi.nlm.nih.gov/34215179/>. We also recapitulate these findings in our study (Fig 1b).

In addition, we performed bacterial quantification to determine the absolute abundance of ARGs in the infant gut. To our knowledge, this is the first report describing the absolute resistome dynamics during early life. Interestingly, in contrast to the gradually decreasing relative ARG abundance, we observed that the absolute ARG load peaks at six months of age.

Concerning the microbiota composition and its role in resistome composition, Xu et al. and Trosvik et al. (<https://pubmed.ncbi.nlm.nih.gov/38882494/>, <https://pubmed.ncbi.nlm.nih.gov/39134593/>), report a correlation between relative *E. coli* abundance and the relative ARG abundance, also previously reported in Lebeaux et al. (<https://pubmed.ncbi.nlm.nih.gov/34215179/>), and further identify resistance genes in *E. coli* genomes. Furthermore, PMC11135851 shows the inverse correlation between the relative abundance of *Bifidobacterium* at genus level, and the prevalence of specific ARGs. Our study confirms the above findings and brings the different aspects together by suggesting a mechanism under which certain early-life bifidobacteria directly might control *E. coli* growth and other early-life ARG-rich pathobionts through the secretion of specific metabolites. In our original manuscript, we showed that ALAs inhibit growth of *E. coli* in an *in vitro* setting, and we have now included further data that support that this inhibition is not limited to *E. coli* but also extends to other ARG-rich bacteria (new Fig. 4g, new Extended Data Fig. 6d and 8a,b).

Although we believe our study provides substantial new insights compared to the above-mentioned studies, we acknowledge the relevance of their findings and have updated our discussion to include them. We edited I333- to ‘We also confirmed that specific bacterial genera were associated with the ARG composition, such as *E. coli*^{17,35,40} and several members of the *Bifidobacterium* genus^{38,41}.’

We understand the reviewer’s comment that it was a limitation in the original manuscript that the *aldh* gene presence was assumed based on phylogeny. This was based on our previous analysis in other cohorts, identifying the *aldh* gene presence in the given taxa (Laursen et al., Nature Microbiol, 2021 (ref. 30, <https://pubmed.ncbi.nlm.nih.gov/34675385/>)). However, to strengthen our claim, we have now performed the analysis to confirm the presence of the *aldh* gene in our cohort specific *Bifidobacterium* MAGs, as suggested. The gene was found in all 7 MAGs that we identified to be associated with the ARG composition (Fig. 2b). This information has been incorporated in the revised manuscript as I245-: ‘Gene analysis showed that all the above-mentioned *Bifidobacterium* MAGs hold the *aldh* gene...’. The *aldh* gene tree has been added as Extended Data Fig. 7a. The approach that was followed to detect the gene in the genomes has been added to the methods section, I548-: ‘Genome annotation was performed for all MAGs with prokka (v 1.14.5)⁶⁸ with default settings. Additionally, 13 genomes of *aldh*+ and *aldh*- bifidobacteria genomes³⁰ were downloaded from NCBI and annotated similar to the MAGs. All genes annotated as lactate dehydrogenase (*ldh*) (EC number 1.1.1.27) were aligned with multiple sequence alignment using clustal-omega (v 1.2.1)⁶⁹ with default settings. A phylogenetic tree was generated based on the alignment with FastTree (v 2.1.11)⁷⁰ and was visualised with iTOL v7⁷¹. All genes that belonged to the same cluster as the *aldh* genes from the annotated reference genomes, and shared the same gene size, were considered as *aldh*, and the MAGs harbouring these genes were considered *aldh*+. All other bifidobacteria strains were designated as *aldh*-.’.

The phylogenetic tree is displayed in new Extended Data Fig. 7a:

Figure text for new Extended Data Fig. 7a: a, Phylogenetic tree of all *ldh* (E.C. 1.1.1.27) genes from 1260 MAGs assembled in this study and of 13 bifidobacteria reference genomes, and zoom in on the part of the tree with *aldh* gene. Genes belonging to ALADDIN MAGs are labeled in blue and those belonging to reference genomes on yellow. The gene size is indicated in parenthesis.

As seen from the below figure, now added as Extended Data Fig. 7b, there is a strong correlation between ALA levels and the abundance of *aldh+* bifidobacteria, whereas *aldh-* bifidobacteria do not correlate with ALA levels.

New Extended Data Fig. 7b:

Figure text for new Extended Data Fig. 7b: b, Spearman correlation of faecal concentrations of the three aromatic lactic acids at 2 months of age vs. relative abundance of the *aldh+* and *aldh-* bifidobacteria as identified based on the above phylogenetic tree. *Aldh-* bifidobacteria define all MAGs belonging to the genus *Bifidobacterium* that do not possess the *aldh* gene based on the phylogenetic tree. Point size represents the P value and the point colour corresponds to the Spearman's correlation coefficient (SCC).

R2-Q2: Besides this limitation, the manuscript is well-written, and the observed results are presented clearly. However, I strongly recommend enhancing the methodological aspects of the resistome and metabolite characterization, as well as providing full transparency in methodology and data reporting. The reported findings seem to be biased due to a lack of a robust methodology.

Answer: We would like to thank the reviewer for the comment about the clarity of our manuscript. We have worked on the different sections that have been pointed out here to further improve transparency in the methodology. Our intention is clearly to make it fully transparent, and we apologize for the lack of sufficient focus on the detail.

Further information on the tools used to annotate and quantify the resistome has been added in I500- and reads "The resistance mechanisms and resistance per antibiotic drug classes were assigned based on the RGI annotation. Genes that were annotated to provide resistance against more than one of the drug classes were characterised as multidrug resistance genes. [...] For both contig- and read-based method, gene abundance was calculated based on RPKM (reads per kilo base million), with the formula: 'reads mapped to gene' * 10^9 divided by 'total reads' * 'gene length'. Absolute ARG abundance was calculated by multiplying the relative ARG abundance (defined as the percentage of an ARG per total gene abundance (ARG RPKM/ 10^4)) with bacteria per gram of faeces.'

The methodology for metabolite characterization has been further clarified in the manuscript as described in the answer to R2-Q19. Additionally, handling and processing of fecal samples prior to metabolite characterization is further clarified in our answer to R2-Q17.

Specific comments:

R2-Q3: The authors choose to use an assembly-based approach to determine the ARGs. However, as low- and even medium-quality reads might be left out in the assembly process, a significant portion of the resistome will not be captured. The limitations of the assembly-based approach have been described earlier (PMID: 39402510, PMID: 36864029). An alternative way to derive the resistome is to map the preprocessed sequencing reads directly to CARD. The validity of conclusions will be more substantial if findings of the alignment- and assembly-based approaches are considered in tandem.

Answer: We would like to thank the reviewer for bringing forward this important point. We applied an assembly-based method to determine ARGs because we were interested in further identifying which microorganisms hold these resistance genes, i.e. the same contigs were used both to identify ARGs and to bin the prokaryotic genomes (MAGs).

However, we recognize the limitations that come from an assembly-based compared to a read-based approach. We have now performed the suggested tandem comparison using the rgi bwt tool from RGI and the CARD database to identify ARGs based on reads (same version as for the contigs to have the most comparable result to the assembly-based output). The read-based approach led to higher ARG abundances for all samples, but within the same range and there was a strong correlation between the assembly- and read-based results for the same sample ($R = 0.97$, $P < 2.2e-16$), as can be seen in the inserted figure below.

Figure text: Pearson correlation of the ARG abundance per sample with the read-based versus the assembly-based method.

Overall, the two approaches show the same ARG abundance shifts over time, both for the relative and absolute abundance (read-based added as Extended Data Fig. 1b and 1e to be compared with original contig-based Fig. 1b and 1c), which further strengthen our findings.

New Extended Data Fig. 1b,e:

Figure text for Extended Data new Fig 1b and 1e: b, Relative abundance of ARGs (RPKM) as defined by the read-based method across time points. e, absolute ARG abundance (relative abundance multiplied by the bacterial load per gram of faeces) as defined by the read-based method across time points.

We have now included the read-based ARG detection data in the manuscript as Extended Data Fig 1b and 1e and we have revised the manuscript in I93- as follows ‘*These findings were verified using a read-mapping ARG identification approach (Extended Data Fig 1b), corroborating previous findings^{12,14,21}, and I108- ‘This change over time was also supported when using the read-based ARG mapping method (Extended Data Fig. 1e)’. The method section has also been updated, I504- ‘In addition to the contig-based approach to identify ARGs described above, a read-based approach was also applied. In brief, trimmed host clean pair reads were mapped to CARD, v3.1.2, using the rgi bwt tool from RGI, v6.0.3. Genes with an average coverage $\geq 80\%$ were considered as positive hits. A strong correlation was identified between the ARG abundance determined using the read- vs. the contig-based approach ($R = 0.97$, $P < 2.2e-16$).’*

R2-Q4: Figure 1b. Can the authors explicitly explain how the relative abundance of ARGs was calculated? Did the ARG diversity comparisons account for sampling depth?

Answer: First, we would like to mention that there unfortunately was a typo in original Fig. 1b, and the y-axis should have indicated values from 0 to 0.75. The relative abundance of the ARGs is based on RPKM (Reads Per Kilobase per Million mapped reads) values which inherently accounts for sampling depth. More precisely, RPKM was calculated with the following formula: Reads mapped to gene $\times 10^9 /$ (total reads in the sample \times gene length). The RPKM value was then divided by 10^4 to display the relative abundance of the ARGs compared to the total gene abundance in the given sample that sums to 100%. We recognize that this choice of data display might not be the easiest to comprehend and have decided to display instead the RPKM values (see revised Fig. 1b below). We have also added an explanation in the methods section on the RPKM calculation, in I500-503: ‘*For both contig- and read-based method, gene abundance was calculated based on RPKM (reads per kilo base million), with the formula: ‘reads mapped to gene’ $\times 10^9$ divided by ‘total reads’ \times ‘gene length’. Absolute ARG abundance was calculated by multiplying the relative ARG abundance (defined as the percentage of an ARG per total gene abundance (ARG RPKM/ 10^4)) with bacteria per gram of faeces.’*

Revised Fig. 1b:

Figure text for revised Fig. 1b: b, Relative abundance of ARGs in Reads Per Kilobase per Million (RPKM) across time points. Samples from 1st week to 6 months were compared with samples from 12 to 60 months with a two-sided Wilcoxon rank-sum test.

R2-Q5: Line 108 – How was multidrug resistance defined?

Answer: We would like to thank the reviewer for bringing to our attention that this has not been clarified in the text. A multidrug resistance gene = every gene annotated as acting against more than one Drug Class according to CARD. We have now added the following sentence: '*Genes that were annotated to provide resistance against more than one of the drug classes were characterized as multidrug resistance genes.*' in the revised manuscript in 1492-493 to clarify the term.

R2-Q6: Line 114 – "This overall suggests that initial seeding of specific ARG-containing bacteria during vaginal delivery is a major contributor to the absolute ARG abundance in the time period between 2 to 6 months of age. " As described in the text now, this statement is not sufficiently backed up by robust statistics and is also in contrast with earlier research (PMID: 36717704). Did the authors control for other factors that significantly influence the gut microbiota and resistome, in particular, the feeding type and antibiotic use? To derive the effect of birth mode on the gut resistome, one could, for example, compare vaginally versus C-section-delivered infants who have the same diet (e.g., all breastfed) and are not exposed to antibiotics. Alternatively, the authors could have used linear mixed-effect modeling to account for the effect of other variables that might be contributing to the effect of birth mode on the gut resistome. The same argument applies to the authors' conclusions on the impact of home births.

Answer: We appreciate the reviewer's relevant comment regarding the need to control for other variables that can influence the gut microbiota and resistome. We have now performed additional analysis to address the above using linear regression models controlling for sex, family lifestyle and feeding mode per time point. Significant differences between the vaginally and caesarean born infants at 2 months, and the home-born vs hospital-born infants at 2 and 6 months were all maintained after controlling for sex, family lifestyle and feeding mode per time point.

Accounting for antibiotics use was a bit more difficult concerning the low number of infants that received antibiotics during the first 6 months; no infants received antibiotics by 2 months of age, and only three children were exposed between 2 and 6 months of age (Extended Data Fig. 3a). For this reason, it has been omitted. Nevertheless, we performed the same analysis by removing the three abx-exposed infants, giving rise to the following values before and after removal of the three infants (see below comparison).

	Estimate	P value	Comment
Birth mode (2m)	-0.34914967	0.04837099	Increased effect after removing the 3 abx-exposed infants
Birth mode (2m, no abx infants)	-0.46135835	0.01939778	
Homebirth (2m)	0.32300405	0.02051109	No change
Homebirth (2m, no abx infants)	0.32300405	0.02051109	
Homebirth (6m)	0.67402348	0.01002495	No change
Homebirth (6m, no abx infants)	0.67402348	0.01002495	

As we observe that removing the three abx-exposed infants results in the same conclusion, we prefer to keep all infants in the analysis.

In response to the changes made here, we have updated the description of the analysis in the methods section (I607-): ‘Effects of environmental variables (delivery mode and home birth) and log₁₀-transformed absolute and relative ARG abundance were assessed by linear regression models using the *lm()* function from the package *stats* (v4.2.1). Models were controlled for sex, family lifestyle, and feeding mode at 2 and 6 months.’ and added new Extended Data Fig. 4a,b (see below), displaying the calculated effect sizes on absolute ARG abundance per time point for birth mode and homebirth.

New Extended Data Fig. 4a,b:

Figure text for Extended Data Fig. 4a & 4b: a,b: Effect size from linear regression models of the log transformed a, absolute ARG abundance and b, relative ARG abundance of the vaginal vs caesarean delivered infants and vaginally delivered infants born at home versus the hospital. Models were controlled for sex, family lifestyle and feeding mode per time point.

Regarding the comment that earlier research has supported the opposite findings, we would like to point out that Busi et al. (ID no. 1 in the table for R1-Q1, <https://pubmed.ncbi.nlm.nih.gov/36717704/>) report

enhanced relative ARG abundance in caesarean vs. vaginally delivered infants at post-natal day 5, while we observe enhanced absolute ARG abundance in vaginally vs. caesarean delivered infants at 2 months of age (Fig. 1e), in homeborn vaginally delivered infants at 2 and 6 months of age (Fig. 1f), and increased transfer of maternal *E. coli* strains to homeborn vaginally delivered infants (Fig. 4b). We did not observe a significant difference in the relative ARG abundance between vaginally and caesarean delivered infants (new Extended Data Fig. 4b). Besides this, the difference between the two studies can also be explained by a) the different cohort characteristics, b) different time points for sampling between the two studies (Busi et al. observe a difference at 5 days, but no differences at 1, 6 and 12 month of age), c) the smaller size of the Busi cohort (20 infants compared to 56 in our study), and d) the differences in antibiotic prophylaxis for caesarean sections that are followed by different countries. Specifically on the last point, we would like to highlight that Sweden does not have a recommendation for antibiotic prophylaxis during C-sections, but 3/7 mothers that delivered with a C-section in our cohort received antibiotics at delivery. This is different from the regimen that is followed in most other countries, as WHO recommends a single dose of prophylactic antibiotics for all women undergoing caesarean section (<https://www.who.int/publications/i/item/9789240028012>). Nevertheless, we did report in our original manuscript that contradicting evidence in literature has been reported (318-: '*Some studies report a higher ARG relative abundance in caesarean section than vaginally delivered infants;...*'). Since a new study has just been published last week (July 2, 2025, ref 38), we have updated the last part of the sentence: '*one identifies it at 1 year of age only¹⁴, and three studies only during the first days of life^{10,37,38}, while there are also reports of higher rates of ARG transfer between mothers and infants in vaginally delivered children³⁹.*

R2-Q7: Figure 2a. Some studies have described opposite trends for resistome α -diversity in infancy (e.g., PMID: 37187112). The reported α -diversity results could be influenced by the assembly-based approach, which will negatively impact the samples at the later time points compared to the earlier time points. At later time points, the community is more complex, but because of the limitations of the assembly, this might result in capturing less of the diversity. It would be helpful to describe to what extent the α -diversity of bacterial microbiota and resistome correlates (e.g., Spearman's correlation and Procrustes analysis) as well as show the richness and the evenness separately. This would give a better sense of what is driving resistome α -diversity and strengthen the conclusions.

Answer: The reviewer is raising a valuable concern on whether the assembly quality of the more diverse samples might drive the observed lower ARG diversity in the later time points. We have tried to address this concern, and we believe the information we provide below confirms the biological relevance of our findings.

First, we would like to establish that we were unable to follow the reviewer's interpretation that the study by Bargheet et al. (<https://pubmed.ncbi.nlm.nih.gov/37187112/>) describes the opposite effect. The group resembling our study group is the FT group, i.e. the full-term born infants, as stated in our methods section, (398: '*Children born pre-term (before gestational week 36) were excluded from the study*'). Although Bargheet et al. do not report statistical differences between the time points in their article, we observe in their Supplementary Figure 8, inserted below, that ARG α -diversity, reflected by the Shannon index, increases from 7d to 28d to 128 days and decreases at 365 days in the FT group, which corroborates our findings.

Supplementary Figure 8 from Bargheet et al. (<https://pubmed.ncbi.nlm.nih.gov/37187112/>)

We observed an inverse correlation ($r = -0.36$, $P < 0.001$) between the ARG alpha-diversity and bacterial alpha-diversity (both reflected by the Shannon index), as can be seen in the inserted figure below. We chose the Shannon index for this as it inherently takes into account both richness and evenness.

Furthermore, as per the reviewer's request, we report below the ARG and species richness based on the Chao index. The ARG richness in absolute ARG numbers is displayed in main Fig. 1a. No correlation is observed between the ARG and bacterial Chao indices, as seen below, indicating that a higher number of detected ARGs cannot be explained by a lower diversity of the sample.

ARG diversity measures based on the read-based ARG prediction confirm the above data. More specifically, we observed a similar inverse correlation ($r = -0.13$, $P < 0.01$) between the bacterial and the ARG Shannon diversity using the read-based ARG detection, as seen in the figure below.

When comparing the values between the two ARG detection methods, a strong correlation was observed ($r = 0.82$, $P < 0.001$), as seen in the figure below.

We believe that the above analyses confirm that the inverse correlation between ARG and bacterial diversity indices reflect a biological effect and not a methodological mistake. This effect might be explained by the lower prevalence and abundance of ARG-rich species in the mature gut, as *E. coli*, *Klebsiella sp.* and *Citrobacter sp.* that are very prominent in the gut of young infants, become less abundant after 12 months of age (see below updated Fig. 3d).

Updated Fig. 3d:

Figure text for updated Fig 3d: d, Absolute abundance of the top 25 MAGs with the highest number of ARGs in faecal samples at the given time points during early life.

R2-Q8: Figure 2c. What is the rationale for using PERMANOVA? In general, PERMANOVA is a test of the effect of parallel variables. In contrast, linear mixed effect models consider the combined impact of all factors, can derive the contribution of each variable while considering other potentially modifying factors, and thus seem more appropriate to use.

Answer: PERMANOVA was used because: a) it is a non-parametric approach, which is preferable when data do not follow a normal distribution as for microbiome data, and b) it works with distance-based matrices, such as the Bray-Curtis dissimilarity distance between samples, being used here. Linear mixed effect models typically assume normal distribution of data and focus on individual data points and not the distance between two objects. PERMANOVA is therefore generally a more suitable method when testing the contribution of different factors on β -diversity indices.

However, we acknowledge that our previous analysis did not account for confounding variables. We have therefore updated our analysis to correct for sex, birth mode and family lifestyle, by performing the PERMANOVA with the following settings:

```
adonis2(ARG_table ~ Sex + Birth mode + Lifestyle + variable, by = 'margin')
```

Mode of birth was corrected for sex and family lifestyle, and family lifestyle was corrected for sex and birth mode.

The updated analysis resulted in significant association of birth mode with the ARG β -diversity for the first two time points, similar to the previous results, while no other variables were found to have a significant association with ARG β -diversity at any time point. We have included an updated Fig. 2c and Extended Data Table 8, and the manuscript has been rephrased to reflect the new findings: (I162-) 'Using a permutational multivariate analysis of variance (PERMANOVA), birth mode (adjusted for multiple testing, sex and family lifestyle) was found to explain a significant fraction of the ARG β -diversity from 1 to 3 weeks of age, while no other variables explained a significant amount of variance in ARG β -diversity (Fig. 2c, Extended Data Table 8)'.

Updated Fig. 2c:

Figure text for updated Fig. 2c: c, perinatal variables associated to ARG β -diversity based on sex, birth mode and family lifestyle adjusted PERMANOVA. Size of the dot depicts the R² and colour depicts the FDR-adjusted P value. FDR adjustment was done per time point.

We have also updated the method section I570- 'Adonis2() was run with the option by = 'margin', accounting for sex, birth mode, and family lifestyle'.

R2-Q9: Line 175: "This implies that strategies to reduce the *E. coli* abundance in the infant gut would have a positive influence on the ARG abundance and potentially reduce the risk of antibiotic resistance spread."  The authors should also consider that an *E. coli* displacement might not be advantageous. There are likely important ecological roles for *E. coli* in the infant gut, including colonization resistance against other Enterobacteriales with pathogenic potential (e.g., PMID: 38096285).

Answer: The reviewer is highlighting an interesting point regarding the ecological role of *E. coli* in the infant gut. The function of *E. coli* in the gut is quite complex and as shown in Spragge et al. (<https://pubmed.ncbi.nlm.nih.gov/38096285/>), it is important in colonization resistance, probably due to direct niche competition with other Enterobacteriaceae among which many are pathogens. Nevertheless, the cited study is performed in an adult animal model using a synthetic gut microbiota and indicates the importance of microbial diversity for providing colonization resistance. It is thus a model system, and we can expect that actual gut ecosystems that are inhabited by more diverse communities should have multiple species that can occupy certain ecological niches, which may indeed support their finding of colonization resistance in adult mice. Additionally, a high *E. coli* abundance in the infant gut has been linked to an asthma-associated microbiome (<https://pubmed.ncbi.nlm.nih.gov/33887206/>).

Based on our findings, we suggest that the *aldh+* bifidobacteria through their ALA metabolites are able to control the abundance of *E. coli* and other ARG-rich clinically relevant pathobionts, and thus, their potential to act as an antibiotic resistance reservoir. However, we do not imply that ALAs can completely eliminate *E. coli* from the gut and, neither do we advocate that complete replacement of *E. coli* is necessary to reduce its role as an AMR reservoir. Rather we suggest that ALAs may be able to better keep the abundance of *E. coli* and other ARG-rich pathobionts in check, not allowing their dominance in the microbial community during the early-life period where *aldh+* bifidobacteria, like *B. infantis*, often expand.

R2-Q10: Figure 4. Many of the ARGs contributing to the measured resistome are likely endogenous genes that do not have much impact on responses to antibiotic treatment in cases of bacterial infection. To focus on the genes that matter most, the authors could highlight the specific bacterial genes and gene classes identified by WHO (and the literature) as the most critical to clinical antibiotic resistance and antibiotic treatment failure, such as carbapenem and vancomycin resistance genes and third generation cephalosporin-resistance genes. Some of these specific ARGs or ARG classes can be carried by multiple Enterobacteriaceae, not just *E. coli*. This would provide direct clinical relevance to the observations.

Answer: We appreciate the reviewer’s valuable suggestion regarding the clinical relevance of the antibiotic resistance genes and have now included this information. A list of clinically relevant antibiotic resistance genes based on the latest WHO Bacterial Priority Pathogen List, 2024 (ref 56) and other literature (ref 57) was compiled, and all ARGs detected in our study were categorized as clinically relevant or not based on the list. Extended-spectrum beta-lactamases (ESBLs) or other genes that confer resistance against carbapenems, 3rd generation cephalosporin, and vancomycin were considered as clinically relevant. Additionally, the list includes the *mcr* gene that confers resistance to colistimethate, a last resort antibiotic, as well as the *tet(X)* genes because they are linked to another last resort antibiotic, tigecycline. Finally, *mecA* was included as it provides resistance against methicillin and methicillin-resistant *S. aureus* is one of the high-risk pathogens according to the WHO Priority list. We have added a table in the supplementary material with all ARGs and their classification (Extended Data Table 2). It is worth noting that many of the identified clinically relevant genes were not detected in our dataset.

We conducted an additional analysis on the abundance of the clinically relevant ARGs in our dataset and observed that their abundance was expectedly lower than the total ARGs, but exhibited a similar age-dependent pattern, as seen below:

New Extended Data Fig. 1f:

Figure text for new Extended Data Fig. 1f: Absolute ARG abundance (contig-based) of the total ARGs or only the clinically relevant ARGs.

We have revised the text I109-: 'Among the ARGs that are annotated as conferring antimicrobial resistance, the subset conferring resistance to clinically relevant antibiotics is more concerning (Extended Data Table 2), such as 3rd generation carbapenems, colistimethate, and tigecycline, which are 'last resort' antibiotics. We further investigated the prevalence and abundance of the subset of clinically relevant ARGs, and observed that the absolute abundance of clinically relevant ARGs followed the same pattern as seen for all ARGs across time, although with larger variation amongst infants until 2 months of age (Extended Data Fig. 1f).'

Additionally, we incorporated information on the number of clinically relevant ARGs that were found per genome and included the analysis on the MAGs with the highest number of clinically relevant ARGs by adding, I182-: 'The majority of MAGs (87.5%) did not possess any clinically relevant ARGs. The highest number was detected in *Klebsiella pneumoniae*, with 7 ARGs, and *Citrobacter freundii*, with 6 ARGs, while *E. coli* contained 4 clinically relevant ARGs (Extended Data Fig. 5a)' adding the Extended Data Fig 5a (see below) and updating Fig 3c.

New Extended Data Fig. 5a:

Figure text for new Extended Data Fig. 5a: Histogram representing the number of clinically relevant ARG in MAGs. MAGs with the highest number of clinically relevant ARGs are named.

Updated Fig. 3c:

Figure text for updated Fig. 3c: c, ARG and virulence gene richness per MAG. The top 25 MAGs with the highest number of ARGs are coloured according to species name. The size of the dots reflects the number of clinically relevant ARGs per MAG.

R2-Q11: Relatedly, can the authors provide a supplemental table listing the detected ARGs and their classification? When evaluating the risk of ARG carriage, the risk level will differ between different ARGs. A list of detected ARGs will reveal whether the genes are clinically relevant or not. The same is valid for virulence genes – since authors do not disclose the detected genes, the conclusions in the manuscript are based only on authors' statements, without any means for verification by others.

Answer: We thank the reviewer for the valuable suggestion. To address this point, we added three additional tables that include further information on the detected ARGs and virulence genes. More specifically:

- New Extended Data Table 2 displays all ARGs detected in the dataset, classified as clinically relevant or not, and the gene family, drug class and resistance mechanism based on CARD.
- New Extended Data Table 9 lists which of the above ARGs were mapped back to a genome.
- New Extended Data Table 11 includes all detected virulence genes and the corresponding genome.

R2-Q12: Figure 4c. Could the authors support the Spearman correlations with a Figure displaying the trajectory of *E. coli* and *Bifidobacterium* spp. abundances over time? The infant feeding type very likely shapes these trajectories; however, this context is missing.

Answer: This is a good idea, and we have now included a new figure, Extended Data Fig. 6a displaying the time-dependent trajectories of the sum of the four *E. coli* MAGs, the sum of the 21 non-*E. coli* ARG-rich taxa and the sum of the four *Bifidobacterium* MAGs showing inverse correlation to *E. coli*. The figure is based on the infants that were fully breastfed at 2 months of age (n=46). We are unfortunately not able to compare based on feeding types, as only 8 infants were partially breastfed at 2 months and 2 were formula fed.

New Extended Data Fig. 6c:

Figure text for new Extended Data Fig. 6c: c, Average relative abundance across time of the sum of the four *E. coli* MAGs, the sum of the four *Bifidobacterium* MAGs (*B. infantis* MAG1, *B. infantis* MAG2, *B. longum* MAG2, *B. longum* MAG3) and the sum of the 21 non-*E. coli* ARG-rich taxa for all infants that were fully breastfed at 2 months (n = 46). Line represents the average and the ribbon represents the standard deviation.

R2-Q13:• Line 220 – In what atmosphere was carried out the experiment with growing *E. coli* isolates in the presence of ALAs? Given the anaerobic conditions of the human gut, using anaerobic conditions would be more appropriate, as the inhibition assay might be affected by differences in bacterial metabolism during aerobic conditions.

Answer: The growth experiments were performed under ambient atmospheric conditions, ca. 760 mm Hg. Both *E. coli*, *Klebsiella pneumoniae*, *Citrobacter freundii* are facultative anaerobes, and can grow both with and without oxygen. We acknowledge that lower concentrations of oxygen may be more appropriate to resemble their growth in an adult intestine, however the growth experiments were intended to reflect the growth conditions of the infant isolates in the early postnatal intestine where levels of butyrate are low/absent, and lactate levels high, which is reported to facilitate epithelial oxygenation (<https://pubmed.ncbi.nlm.nih.gov/23127782/>, <https://pubmed.ncbi.nlm.nih.gov/27078066/>), enabling higher oxygen diffusion into the gut lumen that may benefit the growth of pathobionts (<https://pubmed.ncbi.nlm.nih.gov/27078066/>).

R2-Q14: Extended Data Table 7 □ Can the authors show the complete list of metabolite concentrations per sample and not only the medians?

Answer: We unfortunately have restrictions in terms of sharing personal data. While we acknowledge the importance of data sharing for scientific transparency and reproducibility, our data contains coded data that limits our freedom to share, as the key code can be used to trace back the data to living individuals. According to the Swedish Archive Law, we need to keep the key code for at least 10 years post-publication. The existence of the code means that the data are “pseudonymized personal data”, and The European General Data Protection Regulation (GDPR) act on data protection and privacy imposes restrictions on openly sharing of personal data. Our study has also been conducted in compliance with ethical guidelines, and the participants have not given their consent for their data to be made fully public. Sharing the data openly would not only constitute a GDPR breach but also a breach of the Swedish Ethical Approval law that has approved the research on the terms that personal data are protected as per GDPR and as per the conditions that the participants have agreed to. However, it is possible to inquire for access to the data via the institutional Data Protection Officer that handles requests for data access and ensures compliance with legal and ethical standards.

The data availability statement has been updated in the revised manuscript to provide information on requests to data access (I633-): *‘The non-human metagenomic raw sequencing reads and MAGs for the ALADDIN cohort are deposited under BioProject PRJEB84944. The data on individuals’ nutritional and lifestyle habits, antibiotic exposure, and faecal aromatic lactic acid concentrations are pseudonymized (coded) personal data, prohibited from public sharing. This data is available upon request to johan.alm@ki.se. 16S rRNA gene amplicon sequencing data from the CIG cohort is deposited in the Sequence Read Archive (SRA) under BioProject PRJNA554596’.*

This being said it is possible to provide metabolite levels in a scrambled manner, equal to a sandbox setting, as this makes them non-sensitive. Such an approach may still be valuable from a population perspective. Please let us know if a table with scrambled metabolite data, to support Extended Data Table 12, would be of interest.

R2-Q15: Did the authors detect the *aldh* gene in the metagenomic data, and to which microbial taxa is it* associated? Without these results, the chapter titled "Early life abundance of *E. coli* is kept in check via *aldh*+ bifidobacteria metabolites" and related text is closer to speculation than a factual finding.

Answer: Please refer to our answer to R2-Q1, where we explain in detail how we verified that the involved genomes possess the *aldh* gene.

R2-Q16: Was a bead-beating step included during the DNA extraction? As this step is a source of significant technical variation in microbiome studies, I suggest that authors include this information in the manuscript.

Answer: We agree that bead-beating is essential. The bead tubes type A in the Nucleospin 96 soil kit from Macherey Nagel contain ceramic beads, 0.6-0.8 mm in diameter, and the samples were indeed bead-beated during processing. This has now been specified in the revised version of the manuscript (I460) : '[...] type A containing 0.6-0.8 mm ceramic beads'.

R2-Q17: The reported metabolomics method might be biased by the sample processing. First, the raw fecal samples likely had variations in water content, but it is unclear how this was taken into account. Did the authors do any normalization to make sure the same amount of fecal biomass is processed? Secondly, the method does not report any quenching by an organic solvent during fecal sample collection. Fecal samples that are processed without prior quenching will retain enzymatic activities that can alter metabolite levels.

Answer: We acknowledge that gut bacterial concentrations in faeces may be influenced by stool water content, particularly in cases of very loose stools. However, based on Bristol Stool Scale assessments in this cohort, we did not find water content to be a significant driver of bacterial load overall, as illustrated in the figure below. A significant difference in bacterial load was observed only at the 2-month time point, where children with a stool consistency of type 7 (loose stool) had significantly higher bacterial concentrations per gram of feces compared to those with type 6 stools ($p_{adj} = 0.015$), as determined by a one-way Kruskal-Wallis test, followed by Dunn's post hoc test with Holm correction.

Figure text: Bacterial load was not driven by stool water content. Bacterial load (per gram faeces) at each time point depending on Bristol Stool Scale assessment of the faecal samples. Statistics: One-way Kruskal-Wallis test, followed by Dunn's post hoc test with Holm correction. Horizontal lines indicate the median; box boundaries indicate the interquartile range; whiskers represent values within 1.5x the interquartile range of the first and third quartiles. Dots represent the individual data points.

The bacterial counts are reported per gram of faeces (including water content) rather than per gram of dry biomass. We chose not to normalize faecal samples by dry biomass since analysing the bacteria per gram of faeces with water content more accurately reflects the conditions within the gut – even if samples with higher water content might have had a lower bacterial concentration. Adding to this, dry biomass normalization steps, such as freeze drying of samples to remove water, may introduce additional inter-sample variability.

We did not use organic solvents to quench enzymatic activities in faeces. Faecal samples were frozen at -20°C within 20 minutes after collection, stored at -80°C until sample processing, and kept on ice at all times during sample processing to minimize the influence of retained enzymatic activity upon thawing. We acknowledge that enzymatic activity is retained, and that enzymes can remain active from the time of collection until freezing, and even during freezing. However, using organic solvents to quench the activity increases the risk of 1) introducing a bias due to differences in handling this step participants in-between, 2) disrupting cell membranes which can result in changes in relative abundance of microbial species, and 3) fragmenting or degrading DNA which would reduce the quality and yield of extracted DNA. Additionally, it would also prevent accurate proteomics analyses of the same samples. Since this study and other studies using these faecal samples are combining several omics-readouts from the same samples, we find that freezing rather than quenching by organic solvents is the most compatible approach and best aligns with recommendations for faecal sample processing in prospective cohorts as described in e.g. Sinha et al. 2015 (<https://pubmed.ncbi.nlm.nih.gov/26653756/>). The gold standard of the field has long been immediate freezing at -80°C without any additives or solutions, and brief initial storage at -20°C in cases where cohort participants collect the faecal samples themselves. This remains to be the case today, as evident in e.g. Valdés-Mas et al. 2025 (<https://pubmed.ncbi.nlm.nih.gov/39837331/>).

Moreover, since we here perform inter-individual comparisons between bacterial numbers, ARG load and ALA levels we find that the validity of our findings is relying more on similar handling of samples from all infants than the ability to determine exact metabolite levels. This is further highlighted by the validation of our finding of the inverse association between *E. coli* and *aldh+* bifidobacteria and ALA levels in the independent CIG cohort, as pointed out in our answer to R1-Q1.

R2-Q18: The authors used flow cytometry to quantify bacterial cells in fecal samples. Did they consider that the results might be affected by the formation of aggregates? Using another method for the enumeration of bacterial cells in parallel, such as qPCR, would increase the reliability of the results.

Answer: We did indeed consider this a possibility and investigated this point during our initial gating of samples by visualizing the gated DAPI+ bacteria (viable bacteria, plot to the left in inserted figure below) in an FSC-H vs FSC-A plot, as shown in the middle plot. This shows that no aggregates were identified amongst the gated bacteria population. The population in the top right corner of the plot to the left represents the added count beads encircled in the plot to the right. Count beads are added to define the collected sample volume which we use to calculate the number of bacteria per uL stained solution. Throughout our samples, we did not find aggregates or noticeable duplicates in the bacterial population.

Figure text: Single cell suspensions of gut bacteria did not contain aggregates or noticeable numbers of duplicates. Bacterial single cell suspensions were prepared from faecal samples and stained with DAPI to detect bacterial cells by flow cytometry (Bacteria gate in the leftmost plot), and bacteria were quantified using count beads added just prior to flow cytometry run (Beads gate in the rightmost plot). Gated bacteria did not contain aggregates or noticeable numbers of duplicates, as determined using an FSC-H vs FSC-A plot (middle plot).

To answer the second question on reliability, we have now performed qPCR of total 16S rRNA copies of extracted DNA from newly thawed and processed subsamples collected from a subset of 188 faecal samples with remaining material from the ALADDIN cohort. The qPCR method determines 16S rRNA copy numbers, a number known to vary between bacteria, and includes both viable and dead bacteria, while our flow cytometry procedure for bacterial enumeration includes only viable bacteria.

The comparison of bacterial load/g faeces and 16S rRNA copy numbers/g faeces (see below figure to the left, where the red line represents $X=Y$) shows similar ranges of cells and copy numbers per gram of faeces, despite a weak correlation ($R=0.14$, $P = 0.061$). We observe a slight tendency that more bacteria/g faeces are quantified in samples using flow cytometry as compared to qPCR (plot to the right). Because of the inherent differences between the two methods, we would not expect a perfect correlation, even if samples had been processed simultaneously, and not years apart as in the current case. With this in mind, we hope the reviewer acknowledges our efforts in providing another method to validate the flow cytometry based bacterial count data used for calculation of the absolute ARG and MAG abundance.

Figure text: left, Spearman correlation of the cells/g faeces measured by flow cytometry versus the 16S rRNA gene copy number/g faeces measured by qPCR for the same samples. Right, Variation in total counts/g faeces in the same samples when measured by flow cytometry (FC) and by qPCR.

R2-Q19: Line 365: "the faecal pellet was filtered" – can authors clarify how and why was the fecal pellet filtered?

Answer: The faecal pellet was in fact not filtered. We apologize for this mistake and thank the reviewer for pointing our attention to it. The number of bacterial cells was quantified as described in Eriksen et al. 2024 (ref. 50) with few modifications rather than as described in Moll et al. 2021. Accordingly, we have changed the methodological description and updated the reference, 1440-: 'The number of bacterial cells was quantified as described in Eriksen et al.⁵⁰ with few modifications: Briefly, 100-200 mg faeces was weighed accurately. The homogenized samples, diluted 1:4 in Milli-Q water, were further diluted 2.5-fold in Milli-Q water, and centrifuged at $50 \times g$, 4°C for 15 minutes. Supernatant was transferred to a new tube and centrifuged at $8,000 \times g$, 4°C for 5 minutes. The bacterial pellet was washed twice in sterile PBS + 1% BSA (>98%, Sigma-Aldrich) with centrifugation at $8,000 \times g$, 4°C for 5 minutes, and resuspended in sterile PBS + 1% BSA. An aliquot was diluted 150x in PBS + 1% BSA + 0.01% Tween20 + 1 mM EDTA, mixed with 1 mM DAPI and BD Liquid Counting Beads (BD Biosciences). 200,000 DAPI+ events were acquired on a FACSCanto II (BD Biosciences) flow cytometer. Bacterial counts were defined by gating the bacterial population on SSC-A/Pacific Blue. The number of recorded count beads was used for calculating the recorded sample volume to determine the number of bacteria per μL solution. The bacterial concentration

was multiplied by the dilution factor and divided by the input faecal mass (in grams) to calculate the number of bacteria per g faecal matter.'

R2-Q20: Can authors clarify how anthroposophic lifestyle was defined? What do the authors mean by "anthroposophic" and "non-anthroposophic" families?

Answer: We would like to thank the reviewer for pointing out that the anthroposophic lifestyle was not properly defined in the original manuscript. Families in the ALADDIN cohort are categorized into three lifestyle groups, anthroposophic, partly anthroposophic and non-anthroposophic, based on their choice of Maternal Healthcare Center and parental responses to the questions on a) choice of future type of preschool/school, b) view of lifestyle, and c) the influence of lifestyle on daily life (<https://pubmed.ncbi.nlm.nih.gov/21651566/>). This information has now been included in the manuscript I391-: *'Families were categorized into the lifestyle groups, anthroposophic and non-anthroposophic, based on their choice of Maternal Healthcare Center and parental responses to the questions on a) choice of future type of preschool/school, b) view of lifestyle, and c) the influence of lifestyle on daily life'*. Families with a partly anthroposophic lifestyle were excluded from this study.

R2-Q21: Can the authors include the assembly statistics as supplementary information to document the quality?

Answer: We have now provided the assembly statistics as new Extended Data Table 13.

R2-Q22: As with the resistome and metabolite data, can authors provide the full metadata summarized in Extended Data Table 1? Including all the results according to FAIR data principles will facilitate re-analysis of the dataset and ensure the data integrity.

Answer: Although we recognize the importance of data sharing and would like to adhere to the FAIR data principles, providing detailed information on the participants of the study would be a breach of the Swedish Ethical Approval law and the European General Data Protection Regulation (GDPR). Please refer to our explanation to R2-Q14 for details.

On behalf of all authors,

Susanne Brix

Aug 12, 2025

Dear reviewer #2,

Thank you for your thorough review and feedback on our manuscript NCOMMS-25-03481A 'Temporal dynamics and microbial interactions shaping the gut resistome in early life'.

Below follows our point-by-point response to your comments in blue font and new text additions marked in *italic*, with the line reference referring to the following 'all markup' manuscript file:

Reviewer #2 (Remarks to the Author):

I thank the authors for thoroughly addressing most of my concerns. The manuscript has improved substantially. However, several methodological issues remain.

R2-Q1: The manuscript begins with "... known as Antibiotic Resistance Genes (ARGs), which confer resistance to antibiotics in bacteria, archaea, and eukaryotes." It would be helpful if the authors clarify or provide a reference to support the inclusion of eukaryotes in this context.

Answer: We appreciate the opportunity to clarify the statement in our manuscript "... known as Antibiotic Resistance Genes (ARGs), which confer resistance to antibiotics in bacteria, archaea, and eukaryotes". We formulated this sentence having in mind antimicrobial resistance in fungi, which is an emerging threat to human health (PMID: 35352028). Nevertheless, we recognize that the term antibiotic is used to define compounds targeting prokaryotic cells and the term antimicrobials would be more appropriate when talking about resistance in general. Therefore, we have revisited the sentence and decided to rephrase it focusing specifically on the bacteria as this is also the focus of this study. The sentence now reads 142-43, '*Resistance arises from the expression of specific genes known as Antibiotic Resistance Genes (ARGs), which confer resistance to antibiotics in bacteria*'.

R2-Q2: Regarding the Extended Data Tables, the small text size in the PDF makes them difficult to navigate. Would the authors consider providing these tables in an Excel format to facilitate gene-level exploration and review?

Answer: We agree that the transformation into pdf format during the submission was non-optimal for exploration of these data. We have now provided original Extended Data Table 2, 9, 11 and 13 as Supplementary Data 1-4 in an excel format to enable gene-level exploration and review.

R2-Q3: A key methodological concern relates to the approach used to estimate the absolute abundance of ARGs. The method, which involves multiplying the relative abundance of ARGs by bacterial counts per gram of feces, assumes a uniform distribution of ARGs across bacterial cells. However, the authors themselves show that ARGs are concentrated in specific taxa, such as *E. coli*. Furthermore, the approach assumes a linear relationship between the relative abundance of ARGs and total bacterial counts. For example, if bacteria double in number, ARGs will also double in number. Particularly at 6

months, there is likely an increase in bacterial biomass due to the introduction of solid foods. This raises the possibility that fluctuations in total bacterial load (e.g., around the introduction of solid foods at about 6 months) may not proportionally reflect ARG load, particularly if expanding taxa are ARG-poor. Additionally, could the authors elaborate on the rationale for dividing RPKM values by 10,000 to compute ARG “percentages”? Clarifying this step would help in understanding the conversion from relative to absolute abundances.

Answer: Calculating the ARG relative abundance (RPKM) is an estimate of the proportion of the ARG genes compared to the total number of genes in the bacterial community. Our method does not assume a uniform distribution of ARGs across bacteria but assumes that changes in the ARG relative abundance reflect changes in the abundances of the species that carry them and, thus, by adjusting the relative abundances with the bacterial counts, we account for inter-sample differences in bacterial density and scale ARG relative abundances to more representative levels. Moreover, the method does not assume a linear relationship between relative abundance of ARGs and total bacterial counts. To the contrary, the method assumes that actual abundance of ARGs can be estimated by considering both the relative abundance and the bacterial counts. We agree with the reviewer that introduction of solid foods in principle may result in an increase of the bacterial density. However, we do see an expansion of the community already from 3 weeks of age. Besides, if an expansion is driven by ARG-poor taxa, the relative abundance of ARG-rich taxa will decrease, due to the compositionality, and the relative abundance of the ARGs will be lower for this sample.

Regarding the rationale for dividing RPKM values by 10,000 to compute ARG “percentages”: we recognize that this is not the optimal approach to estimate the absolute ARG abundance and have now adjusted the procedure. The formula for relative abundance in RPKM (reads per kilobase per million mapped reads) is as follows: ‘reads mapped to ARG gene’ / (‘gene length in kilobase pairs’ × ‘reads mapped to all prokaryotic genes’/10⁶). Absolute ARG abundance was calculated as: (RPKM/10⁶) × ‘bacteria per gram of faeces’. The text and the y-axis legend for Fig. 1c,e,f and Sup. Fig. 1e,f has been changed and the explanation in Methods has been adjusted accordingly, 1470-473: “‘reads mapped to ARG gene’ / (‘gene length in kilobase pairs’ × ‘reads mapped to all prokaryotic genes’/10⁶). Absolute ARG abundance was calculated as: (RPKM/10⁶) × ‘bacteria per gram of faeces’”. The calculation has been added to the figure 1 legend. We have further specified in the ‘Identification of microbial resistance genes’ methods section (1442) that the contigs were used to predict prokaryotic genes using Prodigal. Thank you for the opportunity to clarify this.

R2-Q4: On the metabolomics side, it would be valuable to know whether fecal samples were weighed before metabolite quantification. Further clarification on why standards were added at several steps of sample preparation would also enhance reproducibility and understanding of the quantification process. Answer: We appreciate the reviewer’s attention to methodological clarity. Faecal samples were indeed weighed prior to metabolite quantification. To ensure this is clearly communicated, we have revised the first sentence of the methods section titled ‘Quantification of faecal water metabolites’ (1379-380) to

read: *“Aliquots of 100–350 mg from faecal samples collected at 2 and 6 months of age were precisely weighed and diluted 1:4 (w/v) in sterile Milli-Q water and homogenized by vortexing.”*

Regarding the use of internal standards, we would like to clarify that standards were added only once during sample preparation. To avoid any ambiguity, we have revised the relevant sentences (starting at I387 and I395, respectively) to read:

I387, *“Subsequently, 80 µL supernatant was transferred to a new tube in which 20 µL internal standard mix (4 µg/mL isotope-labelled aromatic amino acids: Tyrosine-d4, Phenylalanine-d5, Tryptophan-d5, and Indoleacetic acid-d2) and 240 µL ice-cold acetonitrile were added.”*

L395, *“Targeted UPLC-MS was then employed to quantify the aromatic lactic acids in the faecal water samples using the isotope-labelled aromatic amino acids as internal standards as previously described³⁰.”*

We hope these clarifications improve the transparency and reproducibility of the metabolomics workflow.

R2-Q5: Finally, the interpretation of the influence of birth mode on the resistome could be further strengthened by addressing discrepancies with earlier studies even more thoroughly. For instance, prior reports, including Busi et al. and a new meta-analysis that utilized 3,981 samples from 1,270 infants (PMID: 40048849), have observed higher resistome loads in caesarean-delivered infants under controlled conditions. A more detailed comparison with those findings, and an acknowledgment of potential cohort differences or sample size limitations, would provide helpful context. The fact that the authors in the presented manuscript did not observe a significant difference in the relative abundance of ARGs between vaginally and caesarean-delivered infants (new Extended Data Fig. 4b) supports the statistically underpowered sample size, with small subgroup sizes that limit statistical inference, as highlighted by reviewer 1.

Answer: We recognise that identifying statistically significant correlations between the perinatal parameters and the ARG abundance is dependent on the cohort characteristics, such as sampling size, geography etc. The reviewer points out two studies that detected significantly higher ARG relative abundances in C-section delivered infants compared to those vaginally delivered. We already cite the Busi et al. study (previous version ref. 10, now ref. 11) and have now added the meta-analysis (PMID: 40048849, ref. 38) published in April 2025, after our original submission. Nevertheless, other recently published studies do not report significant differences or report higher abundances in vaginally delivered infants (PMID: 40555747, published in June 2025, ref. 41). The latest study, based on a cohort of 412 infants, highlights that differences among studies cannot be solely attributed to small sample sizes. Based on this, we do not agree that our findings may result from a statistically underpowered sample size; rather, they likely reflect the specific characteristics of the cohort. As we pointed out in the previous response letter, antibiotic exposure during C-section delivery is not as common in Sweden (the

residence country of the families in our study) as in other countries. We appreciate the reviewer's observation regarding cohort differences and agree that these should be clearly acknowledged as influencing the findings, and, to address this even further than in the earlier versions, we have incorporated the following part into the discussion (highlighted part is added) l292-300 '*Some studies report a higher ARG relative abundance in caesarean section than vaginally delivered infants³⁸; one identifies it at 1 year of age only¹⁵, and three studies only during the first days of life^{11,39,40}, while there are also reports of higher ARG relative abundance in vaginally born infants⁴¹ and higher rates of ARG transfer between mothers and infants in vaginally delivered children⁴². No significant differences in ARG relative abundance between birth modes were recorded in our cohort, but we observed a higher ARG absolute abundance in the vaginally born infants at 2 months. These differences between studies could be explained by the different cohort characteristics, e.g. resistome differences between countries due to per capita antibiotic usage⁴³ or country-specific antibiotic prophylaxis for caesarean section*'.

R2-Q6: In conclusion, while the manuscript presents evidence of antibacterial activity by aromatic lactic acids and highlights adh gene clusters in Bifidobacterium MAGs, many of the presented findings have already been discussed in recent studies. The calculation of "ARG absolute abundance" reflects a normalized ARG load per gram of feces rather than a true count of ARGs. The lack of clarification regarding the assumptions underlying the derivation of ARG absolute abundance raises concerns about the validity of the results, making them uncertain and potentially subject to invalidation in the future. Acknowledging the limitations of this method would enhance the robustness and transparency of the findings.

Answer: We acknowledge the reviewer's concern that the 'ARG absolute abundance' reflects a normalized ARG load rather than gene counts. However, we never claimed that we presented gene counts. Our data and figures labels note 'ARG absolute abundance (ARG load per gram feces)' and the way we have calculated these values is explained in the figure legend and Methods section. To our knowledge, this is the first study to use flow cytometry-based bacterial counts to scale ARG abundances, however the method is well established for inferring the abundance of individual taxa based on 16S rRNA amplicon sequencing (PMID: 29143816, PMID: 38123984). Other methods, such as qPCR are more accurate in measuring the exact gene copy numbers but it does so only one gene at a time. Thus, qPCR quantification is not feasible when looking at the total resistome and, even though, adjusting the ARG relative abundance with the bacterial concentration per gram of faeces might not provide gene counts, it does provide additional information in respect to the actual ARG burden. As a result, we do not share the reviewer's concerns regarding the lack of clarification and the invalidity of our findings.

Best regards on behalf of all authors,
Susanne Brix